# Boosting Differentiable Causal Discovery via Adaptive Sample Reweighting

**An Zhang**[1,2]**, Fangfu Liu**[3]**, Wenchang Ma**[2]**, Zhibo Cai**[4]**, Xiang Wang**[*5]**, Tat-seng Chua**[1,2]
[1]Sea-NExT Joint Lab,  [2]National University of Singapore,  [3]Tsinghua University
[4]Renmin University of China,  [5]University of Science and Technology of China
`anzhang@u.nus.edu`, `liuff19@mails.tsinghua.edu.cn`, `e0724290@u.nus.edu`
`caizhibo@ruc.edu.cn`, `xiangwang1223@gmail.com`, `dcscts@nus.edu.sg`

## Abstract

Under stringent model type and variable distribution assumptions, differentiable score-based causal discovery methods learn a directed acyclic graph (DAG) from observational data by evaluating candidate graphs over an average score function. Despite great success in low-dimensional linear systems, it has been observed that these approaches overly exploit easier-to-fit samples, thus inevitably learning spurious edges. Worse still, the common homogeneity assumption can be easily violated, due to the widespread existence of heterogeneous data in the real world, resulting in performance vulnerability when noise distributions vary. We propose a simple yet effective model-agnostic framework to boost causal discovery performance by dynamically learning the adaptive weights for the **Re**weighted **Score** function, **ReScore** for short, where the weights tailor quantitatively to the importance degree of each sample. Intuitively, we leverage the bilevel optimization scheme to alternately train a standard DAG learner and reweight samples — that is, upweight the samples the learner fails to fit and downweight the samples that the learner easily extracts the spurious information from. Extensive experiments on both synthetic and real-world datasets are carried out to validate the effectiveness of ReScore. We observe consistent and significant boosts in structure learning performance. Furthermore, we visualize that ReScore concurrently mitigates the influence of spurious edges and generalizes to heterogeneous data. Finally, we perform the theoretical analysis to guarantee the structure identifiability and the weight adaptive properties of ReScore in linear systems. Our codes are available at https://github.com/anzhang314/ReScore.

## 1 Introduction

Learning causal structure from purely observational data (*i.e.,* causal discovery) is a fundamental but daunting task (Chickering et al., 2004; Shen et al., 2020). It strives to identify causal relationships between variables and encode the conditional independence as a directed acyclic graph (DAG). Differentiable score-based optimization is a crucial enabler of causal discovery (Vowels et al., 2021). Specifically, it is formulated as a continuous constraint optimization problem by minimizing the average score function and a smooth acyclicity constraint. To ensure the structure is fully or partially identifiable (see Section 2), researchers impose stringent restrictions on model parametric family (*e.g.,* linear, additive) and common assumptions of variable distributions (*e.g.,* data homogeneity) (Peters et al., 2014; Ng et al., 2019a). Following this scheme, recent follow-on studies (Kalainathan et al., 2018; Ng et al., 2019b; Zhu et al., 2020; Khemakhem et al., 2021; Yu et al., 2021) extend the formulation to general nonlinear problems by utilizing a variety of deep learning models.

However, upon careful inspections, we spot and justify two unsatisfactory behaviors of the current differentiable score-based methods:

- Differentiable score-based causal discovery is error-prone to learning spurious edges or reverse causal directions between variables, which derails the structure learning accuracy (He et al., 2021;

---

*Xiang Wang is the corresponding author, also with the Institute of Artificial Intelligence, Hefei Comprehensive National Science Center.

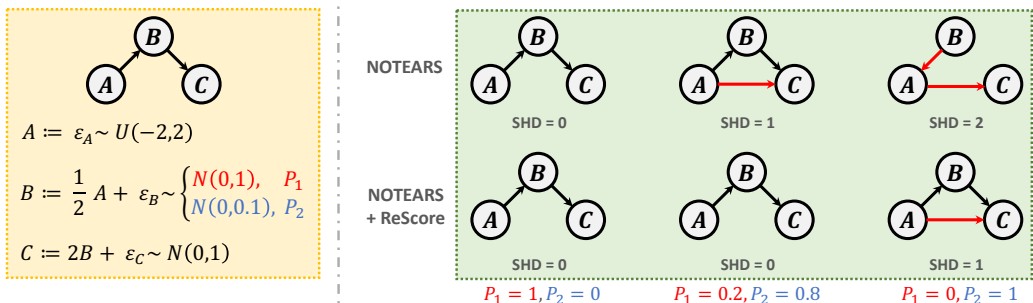

Figure 1: A simple example of basic chain structure that NOTEARS would learn spurious edges while ReScore can help to mitigate the bad influence.

Ng et al., 2022). We substantiate our claim with an illustrative example as shown in Figure 1 (see another example in Appendix D.3.1). We find that even the fundamental chain structure in a linear system is easily misidentified by the state-of-the-art method, NOTEARS (Zheng et al., 2018).

- Despite being appealing in synthetic data, differentiable score-based methods suffer from severe performance degradation when encountering heterogeneous data (Huang et al., 2020; 2019). Considering Figure 1 again, NOTEARS is susceptible to learning redundant causations when the distributions of noise variables vary.

Taking a closer look at this dominant scheme (*i.e.,* optimizing the DAG learner via an average score function under strict assumptions), we ascribe these undesirable behaviors to its inherent limitations:

- The collected datasets naturally include an overwhelming number of easy samples and a small number of informative samples that might contain crucial causation information (Shrivastava et al., 2016). Averagely scoring the samples deprives the discovery process of differentiating sample importance, thus easy samples dominate the learning of DAG. As a result, prevailing score-based techniques fail to learn true causal relationship but instead yield the easier-to-fit spurious edges.

- Noise distribution shifts are inevitable and common in real-world training, as the observations are typically collected at different periods, environments, locations, and so forth (Arjovsky et al., 2019). As a result, the strong assumption of noise homogeneity for differentiable DAG learner is easily violated in real-world data (Peters et al., 2016). A line of works (Ghassami et al., 2018; Wang et al., 2022) dedicated to heterogeneous data can successfully address this issue. However, they often require explicit domain annotations (*i.e.,* ideal partition according to heterogeneity underlying the data) for each sample, which are prohibitively expensive and hard to obtain (Creager et al., 2021), thus further limiting their applicability.

To reshape the optimization scheme and resolve these limitations, we propose to adaptively reweight the samples, which de facto concurrently mitigates the influence of spurious edges and generalizes to heterogeneous data. The core idea is to discover and upweight a set of less-fitted samples that offer additional insight into depicting the causal edges, compared to the samples easily fitted via spurious edges. Focusing more on less-fitted samples enables the DAG learner to effectively generalize to heterogeneous data, especially in real-world scenarios whose samples typically come from disadvantaged domains. However, due to the difficulty of accessing domain annotations, distinguishing such disadvantaged but informative samples and adaptively assigning their weights are challenging.

Towards this end, we present a simple yet effective model-agnostic optimization framework, coined **ReScore**, which automatically learns to reweight the samples and optimize the differentiable DAG learner, without any knowledge of domain annotations. Specifically, we frame the adaptive weights learning and the differentiable DAG learning as a bilevel optimization problem, where the outer-level problem is solved subject to the optimal value of the inner-level problem:

- In the inner loop, the DAG learner is first fixed and evaluated by the reweighted score function to quantify the reliance on easier-to-fit samples, and then the instance-wise weights are adaptively optimized to induce the DAG learner to the worst-case.

- In the outer loop, upon the reweighted observation data where the weights are determined by the inner loop, any differential score-based causal discovery method can be applied to optimize the DAG learner and refine the causal structure.

Benefiting from this optimization scheme, our ReScore has three desirable properties. First, it is a model-agnostic technique that can empower any differentiable score-based causal discovery method. Moreover, we theoretically reveal that the structure identifiability is inherited by ReScore from the original causal discovery method in linear systems (*cf.* Theorem 1). Second, ReScore jointly mitigates the negative effect of spurious edge learning and performance drop in heterogeneous data via auto-learnable adaptive weights. Theoretical analysis in Section 3.3 (*cf.* Theorem 2) validates the oracle adaptive properties of weights. Third, ReScore boosts the causal discovery performance by a large margin. Surprisingly, it performs competitively or even outperforms CD-NOD (Huang et al., 2020) and DICD (Wang et al., 2022), which require domain annotation, on heterogeneous synthetic data and real-world data (*cf.* Section 4.2).

## 2 DIFFERENTIABLE CAUSAL DISCOVERY

We begin by introducing the task formulation of causal discovery and the identifiability issue. We then present the differentiable score-based scheme to optimize the DAG learner.

**Task Formulation.** Causal discovery aims to infer the Structural Causal Model (SCM) (Pearl, 2000; Pearl et al., 2016) from the observational data, which best describes the data generating procedure. Formally, let $\mathbf{X} \in \mathbb{R}^{n \times d}$ be a matrix of observational data, which consists of $n$ independent and identically distributed (i.i.d.) random vectors $X = (X_1, \ldots, X_d) \in \mathbb{R}^d$. Given $\mathbf{X}$, we aim to learn a SCM $(P_X, \mathcal{G})$, which encodes a causal directed acyclic graph (DAG) with a structural equation model (SEM) to reveal the data generation from the distribution of variables $X$. Specifically, we denote the DAG by $\mathcal{G} = (V(\mathcal{G}), E(\mathcal{G}))$, where $V(\mathcal{G})$ is the variable set and $E(\mathcal{G})$ collects the causal directed edges between variables. We present the joint distribution over $X$ as $P_X$, which is Markov *w.r.t.* $\mathcal{G}$. The probability distribution function of $P_X$ is factored as $p(x) = \prod_{i=1}^d P(x_i | x_{pa(i)})$, where $pa(i) = \{j \in V(\mathcal{G}) : X_j \to X_i \in E(\mathcal{G})\}$ is the set of parents of variable $X_i$ in $\mathcal{G}$ and $P(x_i | x_{pa(i)})$ is the conditional probability density function of variable $X_i$ given $X_{pa(i)}$. As a result, the SEM can be formulated as a collection of $d$ structural equations:

$$X_i = f_i(X_{pa(i)}, N_i), \quad i = 1, \cdots, d \tag{1}$$

where $f_i : \mathbb{R}^{|X_{pa(i)}|} \to \mathbb{R}$ can be any linear or nonlinear function, and $N = (N_1, \ldots, N_d)$ are jointly independent noise variables.

**Identifiability Issue.** In general, without further assumption on the SEM (*cf.* Equation 1), it is not possible to uniquely learn the DAG $\mathcal{G}$ by only using the observations of $P_X$. This is the identifiability issue in causal discovery (Lachapelle et al., 2020). Nonetheless, with the assumption of the SEM, the DAG $\mathcal{G}$ is said to be identifiable over $P_X$, if no other SEM can encode the same distribution $P_X$ with a different DAG under the same assumption. To guarantee the identifiability, most prior studies restrict the form of the structural equations to be additive *w.r.t.* to noises, *i.e.,* additive noise models (ANM). Assuming ANM, as long as the structural equations are linear with non-Gaussian errors (Shimizu et al., 2006; Loh & Bühlmann, 2014), linear Gaussian model with equal noise variances (Peters & Bühlmann, 2014), or nonlinear structural equation model with mild conditions (Hoyer et al., 2008; Zhang & Hyvarinen, 2009; Peters et al., 2014), then the DAG $\mathcal{G}$ is identifiable.

**Solution to Causal Discovery.** Prevailing causal discovery approaches roughly fall into two lines: constraint- and score-based methods (Spirtes & Zhang, 2016; Glymour et al., 2019). Specifically, constraint-based methods (Spirtes et al., 1995; Spirtes & Glymour, 1991; Colombo et al., 2012) determine up to the Markov equivalence class of causal graphs, based on conditional independent tests under certain assumptions. Score-based methods (Vowels et al., 2021) evaluate the candidate graphs with a predefined score function and search the DAG space for the optimal graph. Here we focus on the score-based line.

**Score-based Causal Discovery.** With a slight abuse of notation, $\mathcal{G}$ refers to a directed graph in the rest of the paper. Formally, the score-based scheme casts the task of DAG learning as a combinatorial optimization problem:

$$\min_{\mathcal{G}} S(\mathcal{G}; \mathbf{X}) = \mathcal{L}(\mathcal{G}; \mathbf{X}) + \lambda \mathcal{R}_{\text{sparse}}(\mathcal{G}) \quad \text{s.t.} \quad \mathcal{G} \in \text{DAG}, \tag{2}$$

Here this problem consists of two ingredients: the combinatorial acyclicity constraint $\mathcal{G} \in \text{DAG}$ and the score function $S(\mathcal{G}; \mathbf{X})$. The score function composes two terms: (1) the goodness-of-fit

measure $\mathcal{L}(\mathcal{G}; \mathbf{X}) = \frac{1}{n} \sum_{i=1}^{n} l(\mathbf{x}_i, f(\mathbf{x}_i))$, where $l(\mathbf{x}_i, f(\mathbf{x}_i))$ represents the loss of fitting observation $\mathbf{x}_i$; (2) the sparsity regularization $\mathcal{R}_{\text{sparse}}(\mathcal{G})$ stipulating that the total number of edges in $\mathcal{G}$ should be penalized; And $\lambda$ is a hyperparameter controlling the regularization strengths. Next, we will elaborate on the previous implementations of these two major ingredients.

To implement $S(\mathcal{G}; \mathbf{X})$, various approaches have been proposed, such as penalized least-squares loss (Zheng et al., 2020; 2018; Ng et al., 2019b), Evidence Lower Bound (ELBO) (Yu et al., 2019), log-likelihood with complexity regularizers (Kalainathan et al., 2018; Van de Geer & Bühlmann, 2013; Ng et al., 2020), Maximum Mean Discrepancy (MMD) (Goudet et al., 2018), Bayesian Information Criterion (BIC) (Geiger & Heckerman, 1994; Zhu et al., 2020), Bayesian Dirichlet equivalence uniform (BDeu) score (Heckerman et al., 1995), Bayesian Gaussian equivalent (BGe) score (Kuipers et al., 2014), and others (Huang et al., 2018; Bach & Jordan, 2002; Sokolova et al., 2014).

As $\mathcal{G} \in \text{DAG}$ enforces $\mathcal{G}$ to be acyclic, it becomes the main obstacle to the score-based scheme. Prior studies propose various approaches to search in the acyclic space, such as greedy search (Chickering, 2002; Hauser & Bühlmann, 2012), hill-climbing (Gámez et al., 2011; Tsamardinos et al., 2006), dynamic programming (Silander & Myllymäki, 2006; Koivisto & Sood, 2004), A* (Yuan & Malone, 2013), integer linear programming (Jaakkola et al., 2010; Cussens, 2011).

**Differentiable Score-based Optimization.** Different from the aforementioned search approaches, NOTEARS (Zheng et al., 2018) reframes the combinatorial optimization problem as a continuous constrained optimization problem:

$$\min_{\mathcal{G}} \ S(\mathcal{G}; \mathbf{X}) \quad \text{s.t.} \quad H(\mathcal{G}) = 0, \tag{3}$$

where $H(\mathcal{G}) = 0$ is a differentiable equality DAG constraint.

As for the DAG constraint $H(\mathcal{G}) = 0$, the prior effort (Zheng et al., 2018) turns to depict the "DAGness" of $\mathcal{G}$'s adjacency matrix $\mathcal{A}(\mathcal{G}) \in \{0,1\}^{d \times d}$. Specifically, $[\mathcal{A}(\mathcal{G})]_{ij} = 1$ if the causal edge $X_j \rightarrow X_i$ exists in $E(\mathcal{G})$, otherwise $[\mathcal{A}(\mathcal{G})]_{ij} = 0$. Prevailing implementations of DAGness constraints are $H(\mathcal{G}) = \text{Tr}(e^{\mathcal{A} \odot \mathcal{A}}) - d$ (Zheng et al., 2018), $H(\mathcal{G}) = \text{Tr}[(I + \alpha \mathcal{A} \odot \mathcal{A})^d] - d$ (Yu et al., 2019), and others (Wei et al., 2020; Kyono et al., 2020; Bello et al., 2022; Zhu et al., 2021). As a result, this optimization problem in Equation 3 can be further formulated via the augmented Lagrangian method as:

$$\min_{\mathcal{G}} \ S(\mathcal{G}; \mathbf{X}) + \mathcal{P}_{\text{DAG}}(\mathcal{G}), \tag{4}$$

where $\mathcal{P}_{\text{DAG}}(\mathcal{G}) = \alpha H(\mathcal{G}) + \frac{\rho}{2}|H(\mathcal{G})|^2$ is the penalty term enforcing the DAGness on $\mathcal{G}$, and $\rho > 0$ is a penalty parameter and $\alpha$ is the Lagrange multiplier.

## 3 METHODOLOGY OF RESCORE

On the basis of differentiable score-based causal discovery methods, we first devise our ReScore and then present its desirable properties.

### 3.1 BILEVEL FORMULATION OF RESCORE

Aiming to learn the causal structure accurately in practical scenarios, we focus on the observational data that is heterogeneous and contains a large proportion of easy samples. Standard differentiable score-based causal discovery methods apply the average score function on all samples equally, which inherently rely on easy samples to obtain high average goodness-of-fit. As a result, the DAG learner is error-prone to constructing easier-to-fit spurious edges based on the easy samples, while ignoring the causal relationship information maintained in hard samples. Assuming the oracle importance of each sample is known at hand, we can assign distinct weights to different samples and formulate the reweighted score function $S_{\text{w}}(\mathcal{G}; \mathbf{X})$, instead of the average score function:

$$S_{\text{w}}(\mathcal{G}; \mathbf{X}) = \mathcal{L}_{\text{w}}(\mathcal{G}; \mathbf{X}) + \lambda \mathcal{R}_{\text{sparse}}(\mathcal{G}) = \sum_{i=1}^{n} w_i l(\mathbf{x}_i, f(\mathbf{x}_i)) + \lambda \mathcal{R}_{\text{sparse}}(\mathcal{G}), \tag{5}$$

where $\mathbf{w} = (w_1, \ldots, w_n)$ is a sample reweighting vector with length $n$, wherein $w_i$ indicates the importance of the $i$-th observed sample $\mathbf{x}_i$.

However, the oracle sample importance is usually unavailable in real-world scenarios. The problem, hence, comes to how to automatically learn appropriate the sample reweighting vector $\mathbf{w}$. Intuitively, samples easily fitted with spurious edges should contribute less to the DAG learning, while samples that do not hold spurious edges but contain critical information about causal edges should be more importance. We therefore use a simple heuristic of downweighting the easier-to-fit but less informative samples, and upweighting the less-fitted but more informative samples. This inspires us to learn to allocate weights adaptively, with the aim of maximizing the influence of less well-fitted samples and failing the DAG learner. Formally, we cast the overall framework of reweighting samples to boost causal discovery as the following bilevel optimization problem:

$$\min_{\mathcal{G}} \ S_{\mathbf{w}^*}(\mathcal{G}; \mathbf{X}) + \mathcal{P}_{DAG}(\mathcal{G}),$$

$$\text{s.t. } \mathbf{w}^* \in \arg\max_{\mathbf{w} \in \mathbb{C}(\tau)} \ S_{\mathbf{w}}(\mathcal{G}; \mathbf{X}), \tag{6}$$

where $\mathbb{C}(\tau) := \left\{ \mathbf{w} : 0 < \frac{\tau}{n} \leq w_1, \ldots, w_n \leq \frac{1}{\tau n}, \sum_{i=1}^n w_i = 1 \right\}$ for the cutoff threshold $\tau \in (0, 1)$. The deviation of the weight distribution from the uniform distribution is bound by the hyperparameter $\tau$. Clearly, Equation 6 consists of two objectives, where the inner-level objective (*i.e.,* optimize $\mathbf{w}$ by maximizing the reweighted score function) is nested within the outer-level objective (*i.e.,* optimize $\mathcal{G}$ by minimizing the differentiable score-based loss). Solving the outer-level problem should be subject to the optimal value of the inner-level problem.

Now we introduce how to solve this bilevel optimization problem. In the inner loop, we first fix the DAG learner which evaluates the error of each observed sample $\mathbf{x}_i, \forall i \in \{1, \cdots, n\}$, and then maximize the reweighted score function to learn the weight $w_i^*$ correspondingly. In the outer loop, upon the reweighted observations whose weights are determined in the inner loop, we minimize the reweighted score function to optimize the DAG learner. By alternately training the inner and outer loops, the importance of each sample is adaptively estimated based on the DAG learner's error, and in turn gradually guides the DAG learner to perform better on the informative samples. It is worth highlighting that this ReScore scheme can be applied to any differentiable score-based causal discovery method listed in Section 2. The procedure of training ReScore is outlined in Algorithm 1.

Furthermore, our ReScore has the following desirable advantages:

- As shown in Section 3.2, under mild conditions, our ReScore inherits the identifiability property of the original differentiable score-based causal discovery method.

- ReScore is able to generate adaptive weights to observations through the bilevel optimize, so as to distinguish more information samples and fulfill their potentials to guide the DAG learning. This is consistent with our theoretical analysis in Section 3.3 and empirical results in Section 4.2.

- ReScore is widely applicable to various types of data and models. In other words, it is model-agnostic and can effectively handle heterogeneous data without knowing the domain annotations in advance. Detailed ReScore performance can be found in Section 4.

## 3.2 THEORETICAL ANALYSIS ON IDENTIFIABILITY

The graph identifiability issue is the primary challenge hindering the development of structure learning. As an optimization framework, the most desired property of ReScore is the capacity to ensure graph identifiability and substantially boost the performance of the differentiable score-based DAG learner. We develop Theorem 1 that guarantees the DAG identifiability when using ReScore.

Rendering a DAG theoretically identifiable requires three standard steps (Peters et al., 2014; Zheng et al., 2020; Ng et al., 2022): (1) assuming the particular restricted family of functions and data distributions of SEM in Equation 1; (2) theoretically proving the identifiability of SEM; and (3) developing an optimization algorithm with a predefined score function and showing that learned DAG asymptotically converges to the ground-truth DAG. Clearly, ReScore naturally inherits the original identifiability of a specific SEM as stated in Section 2. Consequently, the key concern lies on the third step — whether the DAG learned by our new optimization framework with the reweighted score function $S_{\mathbf{w}}(\mathcal{G}; \mathbf{X})$ can asymptotically converge to the ground-truth DAG. To address this, we present the following theorem. Specifically, it demonstrates that, by guaranteeing the equivalence of optimization problems (Equation 2 and Equation 6) in linear systems, the bounded weights will not affect the consistency results in identifiability analysis. See detailed proof in Appendix C.1.

**Theorem 1.** *Suppose the SEM in Equation 1 is linear and the size of observational data $\boldsymbol{X}$ is $n$. As the data size increases,* i.e., $n \to \infty$,

$$\arg\min_{\mathcal{G}} \left\{ S_{\mathbf{w}}(\mathcal{G}; \boldsymbol{X}) + \mathcal{P}_{DAG}(\mathcal{G}) \right\} - \arg\min_{\mathcal{G}} \left\{ S(\mathcal{G}; \boldsymbol{X}) + \mathcal{P}_{DAG}(\mathcal{G}) \right\} \xrightarrow{a.s.} \mathbf{0}$$

*in the following cases:*

    *a. Using the least-squares loss $\mathcal{L}(\mathcal{G}; \boldsymbol{X}) = \frac{1}{2n} \| \boldsymbol{X} - f(\boldsymbol{X}) \|_F^2$;*

    *b. Using the negative log-likelihood loss with standard Gaussian noise.*

**Remark:** The identifiability property of ReScore with two most common score functions, namely least-square loss and negative log-likelihood loss, is proved in Theorem 1. Similar conclusions can be easily derived for other loss functions, which we will explore in future work.

### 3.3 ORACLE PROPERTY OF ADAPTIVE WEIGHTS

Our ReScore suggests assigning varying degrees of importance to different observational samples. At its core is the simple yet effective heuristic: the less-fitted samples are more important than the easier-to-fit samples, as they do not hold spurious edges but contain critical information about the causal edges. Hence, mining hard-to-learn causation information is promising to help DAG learners mitigate the negative influence of spurious edges. The following theorem shows the adaptiveness property of ReScore, *i.e.,* instead of equally treating all samples, ReScore tends to upweight the importance of hard but informative samples while downweighting the reliance on easier-to-fit samples.

**Theorem 2.** *Suppose that in the optimization phase, the $i$-th observation has a larger error than the $j$-th observation in the sense that $l(\boldsymbol{x}_i, f(\boldsymbol{x}_i)) > l(\boldsymbol{x}_j, f(\boldsymbol{x}_j))$, where $i, j \in \{1, \ldots, n\}$. Then,*

$$w_i^* \geq w_j^*,$$

*where $w_i^*, w_i^*$ are the optimal weights in Equation 6. The equality holds if and only if $w_i^* = w_j^* = \frac{\tau}{n}$ or $w_i^* = w_j^* = \frac{1}{\tau n}$.*

See Appendix C.2 for the detailed proof. It is simple to infer that, following the inner loop that maximizes the reweighted score function $S_{\mathrm{w}}(\mathcal{G}; \mathbf{X})$, the observations are ranked by learned adaptive weights $\mathbf{w}^*$. That is, one observation equipped with a higher weight will have a greater impact on the subsequent outer loop to dominate the DAG learning.

## 4 EXPERIMENTS

We aim to answer the following research questions:

- **RQ1:** As a model-agnostic framework, can ReScore widely strengthen the differentiable score-based causal discovery baselines?

- **RQ2:** How does ReScore perform when noise distribution varies? Can ReScore effectively learn the adaptive weights that successfully identify the important samples?

**Baselines.** To answer the first question (RQ1), we implement various backbone models including NOTEARS (Zheng et al., 2018) and GOLEM (Ng et al., 2020) in linear systems, and NOTEARS-MLP (Zheng et al., 2020), and GraN-DAG (Lachapelle et al., 2020) in nonlinear settings. To answer the second question (RQ2), we compare GOLEM+ReScore, NOTEARS-MLP+ReScore to a SOTA baseline CD-NOD (Huang et al., 2020) and a recently proposed approach DICD (Wang et al., 2022), which both require the ground-truth domain annotation. For a comprehensive comparison, extensive experiments are conducted on both homogeneous and heterogeneous synthetic datasets as well as a real-world benchmark dataset, *i.e.,* Sachs (Sachs et al., 2005). In Sachs, GES (Chickering, 2002), a benchmark discrete score-based causal discovery method, is also considered. A detailed description of the employed baselines can be found in Appendix D.1.

**Evaluation Metrics.** To evaluate the quality of structure learning, four metrics are reported: True Positive Rate (TPR), False Discovery Rate (FDR), Structural Hamming Distance (SHD), and Structural Intervention Distance (SID) (Peters & Bühlmann, 2015), averaged over ten random trails.

Table 1: Results for ER graphs of 10 nodes on linear and nonlinear synthetic datasets.

| | ER2 | | | | ER4 | | | |
|---|---|---|---|---|---|---|---|---|
| | TPR ↑ | FDR ↓ | SHD ↓ | SID ↓ | TPR ↑ | FDR ↓ | SHD ↓ | SID ↓ |
| Random | $0.08_{\pm0.07}$ | $0.93_{\pm0.18}$ | $33.2_{\pm7.3}$ | $95.6_{\pm12.2}$ | $0.09_{\pm0.17}$ | $0.93_{\pm0.09}$ | $52.3_{\pm16.7}$ | $80.3_{\pm17.7}$ |
| NOTEARS | $0.85_{\pm0.09}$ | $\mathbf{0.07}_{\pm0.07}$ | $5.8_{\pm2.2}$ | $20.8_{\pm5.2}$ | $0.79_{\pm0.11}$ | $0.09_{\pm0.05}$ | $10.0_{\pm5.2}$ | $25.8_{\pm9.9}$ |
| + ReScore | $\mathbf{0.89}_{\pm0.07}{}^{+5\%}$ | $0.08_{\pm0.09}{}^{-12\%}$ | $\mathbf{4.6}_{\pm2.3}{}^{+26\%}$ | $12.8_{\pm7.0}{}^{+63\%}$ | $\mathbf{0.85}_{\pm0.04}{}^{+8\%}$ | $\mathbf{0.05}_{\pm0.04}{}^{+57\%}$ | $\mathbf{7.2}_{\pm1.9}{}^{+39\%}$ | $\mathbf{24.2}_{\pm8.4}{}^{+7\%}$ |
| GOLEM | $0.87_{\pm0.06}$ | $0.22_{\pm0.11}$ | $6.5_{\pm3.4}$ | $13.0_{\pm6.7}$ | $0.63_{\pm0.03}$ | $0.16_{\pm0.03}$ | $17.2_{\pm1.3}$ | $48.0_{\pm13.3}$ |
| + ReScore | $0.88_{\pm0.06}{}^{+1\%}$ | $0.21_{\pm0.11}{}^{+2\%}$ | $6.0_{\pm3.4}{}^{+8\%}$ | $\mathbf{12.4}_{\pm6.3}{}^{+5\%}$ | $0.66_{\pm0.04}{}^{+5\%}$ | $0.17_{\pm0.01}{}^{-5\%}$ | $16.2_{\pm1.0}{}^{+6\%}$ | $46.7_{\pm13.3}{}^{+3\%}$ |
| NOTEARS-MLP | $0.76_{\pm0.17}$ | $0.14_{\pm0.09}$ | $7.0_{\pm3.5}$ | $17.9_{\pm10.0}$ | $0.83_{\pm0.05}$ | $0.21_{\pm0.04}$ | $10.9_{\pm1.9}$ | $28.6_{\pm12.0}$ |
| + ReScore | $0.73_{\pm0.07}{}^{-4\%}$ | $0.10_{\pm0.09}{}^{+37\%}$ | $6.8_{\pm2.9}{}^{+3\%}$ | $20.3_{\pm9.7}{}^{-11\%}$ | $0.94_{\pm0.06}{}^{+14\%}$ | $0.15_{\pm0.06}{}^{+44\%}$ | $6.80_{\pm2.7}{}^{+60\%}$ | $8.80_{\pm12.4}{}^{+225\%}$ |
| GraN-DAG | $0.88_{\pm0.06}$ | $0.02_{\pm0.03}$ | $2.7_{\pm1.6}$ | $8.70_{\pm4.8}$ | $0.98_{\pm0.02}$ | $0.12_{\pm0.03}$ | $5.4_{\pm1.1}$ | $3.70_{\pm4.71}$ |
| + ReScore | $\mathbf{0.90}_{\pm0.05}{}^{+2\%}$ | $\mathbf{0.01}_{\pm0.03}{}^{+35\%}$ | $\mathbf{2.4}_{\pm1.1}{}^{+13\%}$ | $\mathbf{7.20}_{\pm3.0}{}^{+21\%}$ | $\mathbf{0.99}_{\pm0.01}{}^{+1\%}$ | $\mathbf{0.11}_{\pm0.01}{}^{+12\%}$ | $\mathbf{4.80}_{\pm0.6}{}^{+13\%}$ | $\mathbf{0.50}_{\pm0.81}{}^{+640\%}$ |

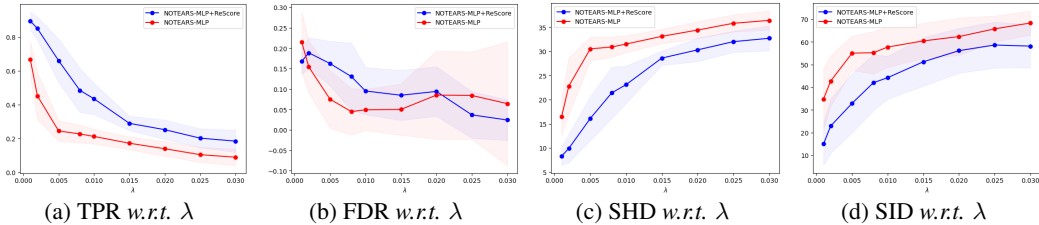

| (a) TPR *w.r.t.* $\lambda$ | (b) FDR *w.r.t.* $\lambda$ | (c) SHD *w.r.t.* $\lambda$ | (d) SID *w.r.t.* $\lambda$ |
|---|---|---|---|

Figure 2: Performance comparison between NOTEARS-MLP and ReScore on ER4 graphs of 10 nodes on nonlinear synthetic datasets. The hyperparameter $\lambda$ defined in Equation 2 refers to the graph sparsity. See more results in Appendix D.4

## 4.1 OVERALL PERFORMANCE COMPARISON ON SYNTHETIC DATA (RQ1)

**Simulations.** The generating data differs along three dimensions: number of nodes, the degree of edge sparsity, and the type of graph. Two well-known graph sampling models, namely Erdos-Renyi (ER) and scale-free (SF) (Barabási & Albert, 1999) are considered with $kd$ expected edges (denoted as ER$k$ or SF$k$) and $d = \{10, 20, 50\}$ nodes. Specifically, in linear settings, similar to (Zheng et al., 2018; Gao et al., 2021), the coefficients are assigned following $U(-2, -0.5) \cup U(0.5, 2)$ with additive standard Gaussian noise. In nonlinear settings, following (Zheng et al., 2020), the ground-truth SEM in Equation 1 is generated under the Gaussian process (GP) with radial basis function kernel of bandwidth one, where $f_i(\cdot)$ is additive noise models with $N_i$ as an i.i.d. random variable following standard normal distribution. Both of these settings are known to be fully identifiable (Peters & Bühlmann, 2014; Peters et al., 2014). For each graph, 10 data sets of 2,000 samples are generated and the mean and standard deviations of the metrics are reported for a fair comparison.

**Results.** Tables 1, 9 and Tables in Appendix D.4 report the empirical results on both linear and nonlinear synthetic data. The error bars depict the standard deviation across datasets over ten trails. The red and blue percentages separately refer to the increase and decrease of ReScore relative to the original score-based methods in each metric. The best performing methods are bold. We find that:

- **ReScore consistently and significantly strengthens the score-based methods for structure learning across all datasets.** In particular, it achieves substantial gains over the state-of-the-art baselines by around 3% to 60% in terms of SHD, revealing a lower number of missing, falsely detected, and reversed edges. We attribute the improvements to the dynamically learnable adaptive weights, which boost the quality of score-based DAG learners. With a closer look at the TPR and FDR, ReScore typically lowers FDR by eliminating spurious edges and enhances TPR by actively identifying more correct edges. This clearly demonstrates that ReScore effectively filters and upweights the more informative samples to better extract the causal relationship. Figure 2 also illustrates the clear trend that ReScore is excelling over NOTEARS-MLP as the sparsity penalty climbs. Additionally, as Table 7 indicates, ReScore only adds a negligible amount of computational complexity as compared to the backbone score-based DAG learners.

- **Score-based causal discovery baselines suffer from a severe performance drop on high-dimensional dense graph data.** Despite the advances, beyond linear, NOTEARS-MLP and GraN-DAG fail to scale to more than 50 nodes in SF4 and ER4 graphs, mainly due to difficulties in enforcing acyclicity in high-dimensional dense graph data (Varando, 2020; Lippe et al., 2022). Specifically, the TPR of GraN-DAG and NOTEARS-MLP in SF4 of 50 nodes is lower than 0.2, which indicates that they are not even able to accurately detect 40 edges out of 200 ground-truth edges. ReScore, as an optimization framework, relies heavily on the performance of the score-

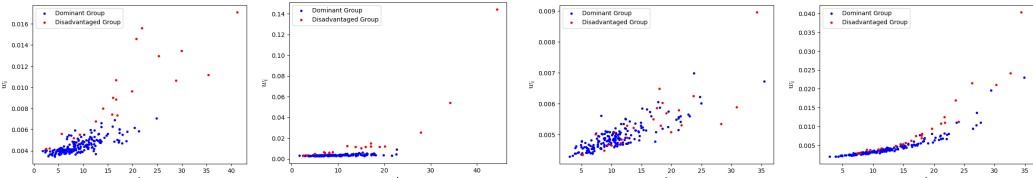

(a) Weights on linear data at the 1st and last epochs (b) Weights on nonlinear data at the 1st and last epochs

Figure 3: Illustration of adaptive weights learned by ReScore *w.r.t.* sample loss on both linear and nonlinear synthetic data. For each dataset, the left and right plots refer to the distribution of adaptive weights at the first and last epochs in the outer loop, respectively (*i.e.*, the value of $\mathbf{w}^*$, when $k_1 = 0$ and $k_1 = K_{outer}$ in Algorithm 1, respectively). The disadvantaged but more informative samples are represented by the red dots. The dominant and easy samples, in contrast, are in blue.

based backbone model. When the backbone model fails to infer DAG on its own as the number of nodes and edge density increase, adding ReScore will not be able to enhance the performance.

## 4.2 PERFORMANCE ON HETEROGENEOUS DATA (RQ2)

### 4.2.1 EVALUATION ON SYNTHETIC HETEROGENEOUS DATA

**Motivations.** It is commonplace to encounter heterogeneous data in real-world applications, of which the underlying causal generating process remain stable but the noise distribution may vary. Specific DAG learners designed for heterogeneous data are prone to assume strict conditions and require the knowledge of group annotation for each sample. Group annotations, however, are extremely costly and challenging to obtain. We conjecture that a robust DAG learner is able to successfully handle heterogeneous data without the information of group annotation.

**Simulations.** Synthetic heterogeneous data in both linear and nonlinear settings ($n = 1000$, $d = 20$, ER2) containing two distinct groups are also considered. 10% of observations come from the disadvantaged group, where half of the noise variables $N_i$ defined in Equation 1 follow $\mathcal{N}(0, 1)$ and the remaining half of noise variables follow $\mathcal{N}(0, 0.1)$. 90% of the observations, in contrast, are generated from the dominant group where the scales of noise variables are flipped.

**Results.** To evaluate whether ReScore can handle heterogeneous data without requiring the group annotation by automatically identifying and upweighting informative samples, we compare baseline+ReScore to CD-NOD and DICD, two SOTA causal discovery approaches that rely on group annotations and are developed for heterogeneous data. Additionally, a non-adaptive reweighting method called baseline+IPS is taken into account, in which sample weights are inversely proportional to group sizes. Specifically, we divide the whole observations into two subgroups. Obviously, a single sample from the disadvantaged group is undoubtedly more informative than a sample from the dominant group, as it offers additional insight to depict the causal edges.

As Figure 3 shows, dots of different colours are mixed and scattered at the beginning of the training. After multiple iterations of training in inner and outer loops, the red dots from the disadvantaged group are gradually iden-

Table 2: Results on heterogeneous data.

| Linear | TPR↑ | FDR↓ | SHD↓ | Nonlinear | TPR↑ | FDR↓ | SHD↓ |
|---|---|---|---|---|---|---|---|
| GOLEM | 0.79 | 0.33 | 18.7 | NOTEARS-MLP | 0.62 | 0.36 | 25.8 |
| + IPS | 0.65 | 0.19 | 18.6 | + IPS | 0.35 | 0.21 | 28.7 |
| + ReScore | 0.81 | 0.24 | 16.4 | + ReScore | 0.63 | 0.32 | 23.8 |
| CD-NOD | 0.51 | 0.17 | 24.1 | CN-NOD | 0.60 | 0.29 | 26.0 |
| DICD | 0.82 | 0.28 | 16.7 | DICD | 0.50 | 0.24 | 23.5 |

tified and assigned to relatively larger weights as compared to those blue dots with the same measure-of-fitness. This illustrates the effectiveness of ReScore and further offers insight into the reason for its performance improvements when handling heterogeneous data. Overall, all figures show clear positive trends, *i.e.,* the underrepresented samples tend to learn bigger weights. These results validate the property of adaptive weights in Theorem 2.

Table 2 indicates that ReScore drives impressive performance breakthroughs in heterogeneous data, achieving competitive or even lower SHD without group annotations compared to CD-NOD and DICD recognized as the lower bound. Specifically, both GOLEM and NOTEARS-MLP are struggling from notorious performance drop when homogeneity assumption is invalidated, and posing hurdle from being scaled up to real-world large-scale applications. We ascribe this hurdle to blindly scoring the observational samples evenly, rather than distilling the crucial group information from

distribution shift of noise variables. To better highlight the significance of the adaptive property, we also take Baseline+IPS into account, which views the ratio of group size as the propensity score and exploits its inverse to re-weight each sample's loss. Baseline+IPS suffers from severe performance drops in terms of TPR, revealing the limitation of fixed weights. In stark contrast, benefiting from adaptive weights, ReScore can even extract group information from heterogeneous data that accomplish more profound causation understanding, leading to higher DAG learning quality. This validates that ReScore endows the backbone score-based DAG learner with better robustness against the heterogeneous data and alleviates the negative influence of spurious edges.

### 4.2.2 Evaluations on Real Heterogeneous Data.

Sachs (Sachs et al., 2005) contains the measurement of multiple phosphorylated protein and phospholipid components simultaneously in a large number of individual primary human immune system cells. In Sachs, nine different perturbation conditions are applied to sets of individual cells, each of which administers certain reagents to the cells. With the annotations of perturbation conditions, we consider the Sachs as real-world heterogeneous data (Mooij et al., 2020). We train baselines on 7,466 samples, where the ground-truth graph (11 nodes and 17 edges) is widely accepted by the biological community.

Table 3: The performance comparison on Sachs dataset.

|  | TPR ↑ | FDR ↓ | SHD ↓ | SID ↓ | #Predicted Edges |
|---|---|---|---|---|---|
| **Random** | 0.076 | 0.899 | 23 | 63 | 22 |
| **GOLEM** | 0.176 | 0.026 | 15 | 53 | 4 |
| **+ ReScore** | 0.294 | 0.063 | **14** | 49 | 6 |
| **NOTEARS-MLP** | 0.412 | 0.632 | 16 | 45 | 19 |
| **+ ReScore** | 0.412 | 0.500 | **13** | 43 | 14 |
| **GraN-DAG** | 0.294 | 0.643 | 16 | 60 | 14 |
| **+ ReScore** | 0.353 | 0.600 | **15** | 58 | 15 |
| **GES** | 0.294 | 0.853 | 31 | 54 | 34 |
| **+ ReScore** | 0.588 | 0.722 | **28** | 50 | 36 |
| **CD-NOD** | 0.588 | 0.444 | 15 | - | 18 |

As Table 3 illustrates, ReScore steadily and prominently boosts all baselines, including both differentiable and discrete score-based causal discovery approaches *w.r.t.* SHD and SID metrics. This clearly shows the effectiveness of ReScore to better mitigate the reliance on easier-to-fit samples. With a closer look at the TPR and FDR, baseline+ReScore surpasses the state-of-the-art corresponding baseline by a large margin in most cases, indicating that ReScore can help successfully predict more correct edges and fewer false edges. Remarkably, compared to CD-NOD, which is designed for heterogeneous data and utilizes the annotations as prior knowledge, GES+ReScore obtains competitive TPR without using ground-truth annotations. Moreover, GraN-DAG+ReScore can reach the same SHD as CD-NOD when 15 and 18 edges are predicted, respectively. These findings validate the potential of ReScore as a promising research direction for enhancing the generalization and accuracy of DAG learning methods when dealing with real-world data.

## 5 Conclusion

Today's differentiable score-based causal discovery approaches are still far from being able to accurately detect the causal structures, despite their great success on synthetic linear data. In this paper, we proposed ReScore, a simple-yet-effective model-agnostic optimization framework that simultaneously eliminates spurious edge learning and generalizes to heterogeneous data by utilizing learnable adaptive weights. Grounded by theoretical proof and empirical visualization studies, ReScore successfully identifies the informative samples and yields a consistent and significant boost in DAG learning. Extensive experiments verify that the remarkable improvement of ReScore on a variety of synthetic and real-world datasets indeed comes from adaptive weights.

Two aspects of ReScore's limitations will be covered in subsequent works. First, the performance of ReScore is highly related to the causal discovery backbone models, which leads to minor improvements when the backbone methods fail. Second, having empirically explored the sensitivity to pure noise samples in D.3.2, we will theoretically analyze and further enhance the robustness of ReScore against these noises. It is expected to substantially improve the DAG learning quality, as well as distinguish true informative samples from pure noise samples. We believe that ReScore provides a promising research direction to diagnose the performance degradation for nonlinear and heterogeneous data in the structure learning challenge and will inspire more works in the future.

ACKNOWLEDGMENTS

This research is supported by the Sea-NExT Joint Lab, the National Natural Science Foundation of China (9227010114), and CCCD Key Lab of the Ministry of Culture and Tourism.

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

# A    RELATED WORK

**Differentiable score-based causal discovery methods.** Learning the directed acyclic graph (DAG) from purely observational data is challenging, owing mainly to the intractable combinatorial nature of acyclic graph space. A recent breakthrough, NOTEARS (Zheng et al., 2018), formulates the discrete DAG constraint into a continuous equality constraint, resulting in a differentiable score-based optimization problem. Recent subsequent works extends the formulation to deal with nonlinear problems by using a variety of deep learning models, such as neural networks (NOTEARS+ (Zheng et al., 2020), GraN-DAG (Lachapelle et al., 2020), CASTLE (Kyono et al., 2020), MCSL (Ng et al., 2019a), DARING (He et al., 2021)), generative autoencoder (CGNN (Goudet et al., 2018), Causal-VAE (Yang et al., 2021), ICL (Wang et al., 2020), DAG-GAN (Gao et al., 2021)), graph neural network (DAG-GNN (Gao et al., 2021), GAE (Ng et al., 2019b)), generative adversarial network (SAM (Kalainathan et al., 2018), ICL (Wang et al., 2020)), and reinforcement learning (RL-BIC (Zhu et al., 2020)).

**Multi-domain causal structure learning.** Most multi-domain causal structure learning methods are constraint-based and have diverse definition of domains. In our paper, the multi-domain or multi-group refers to heterogeneous data whose underlying causal generating process remain stable but the distributions of noise variables may vary. In literature, our definition of multi-domain is consistent with MC (Ghassami et al., 2018), CD-NOD (Huang et al., 2020), LRE (Ghassami et al., 2017), DICD (Wang et al., 2022), and others (Peters et al., 2016). In addition to the strict restriction of knowing the domain annotation in advance, the majority of structure learning models dedicated to heterogeneous data exhibit limited applicability, due to linear case assumption (Ghassami et al., 2018; 2017), causal direction identification only (Huang et al., 2019; Cai et al., 2020), and time-consuming (Huang et al., 2020).

# B    ALGORITHM OF RESCORE

---

**Algorithm 1** ReScore Algorithm for Differentiable Score-based Causal Discovery

---

**Input:** observational data $\mathcal{D}$: $\{\mathbf{x}_i : i = 1, 2, ..., n\}$, DAG learner parameters $\theta_{\mathcal{G}}$, reweighting model parameters $\theta_w$, cutoff threshold $\tau$, epoch to start reweighting $K_{reweight}$, maximum epoch in the inner loop $K_{inner}$, maximum epoch in the outer loop $K_{outer}$
**Initialize:** initialize $\theta_w$ to uniformly output $\frac{1}{n}$, $k_1 = 0$, $k_2 = 0$
**for** $k_1 \leq K_{outer}$ **do**
    Fix reweighting model parameters $\theta_w$
    Calculate $\mathbf{w}^*$ by applying threshold $[\frac{\tau}{n}, \frac{1}{n\tau}]$
    Optimize $\theta_{\mathcal{G}}$ by minimizing $S_{\mathbf{w}^*}(\mathcal{G}; \mathbf{X}) + \mathcal{P}_{DAG}(\mathcal{G})$    # Outer optimization in Equation 6
    **if** $k_1 \geq k_{reweight}$ **then**
        **for** $k_2 \leq K_{inner}$ **do**
            Fix the DAG learner's parameters $\theta_{\mathcal{G}}$
            Get $\mathbf{w}$ from $\theta_w$ by applying threshold $[\frac{\tau}{n}, \frac{1}{n\tau}]$
            Optimize $\theta_w$ by maximizing $S_{\mathbf{w}}(\mathcal{G}; \mathbf{X})$    # Inner optimization in Equation 6
            $k_2 \leftarrow k_2 + 1$
        **end for**
        $k_1 \leftarrow k_1 + 1$
        $k_2 \leftarrow 0$
    **end if**
**end for**
**return** predicted $\mathcal{G}$ from DAG learner

---

## C  IN-DEPTH ANALYSIS OF RESCORE

### C.1  PROOF OF THEOREM 1

**Theorem 1.** *Suppose the SEM in Equation 1 is linear and the size of observational data $X$ is $n$. As the data size increases, i.e., $n \to \infty$,*

$$\arg\min_{\mathcal{G}} \left\{ S_{\boldsymbol{w}}(\mathcal{G}; \boldsymbol{X}) + \mathcal{P}_{DAG}(\mathcal{G}) \right\} - \arg\min_{\mathcal{G}} \left\{ S(\mathcal{G}; \boldsymbol{X}) + \mathcal{P}_{DAG}(\mathcal{G}) \right\} \xrightarrow{a.s.} \boldsymbol{0}$$

*in the following cases:*

    *a. Using the least-squares loss $\mathcal{L}(\mathcal{G}; \boldsymbol{X}) = \frac{1}{2n} \|\boldsymbol{X} - f(\boldsymbol{X})\|_F^2$;*

    *b. Using the negative log-likelihood loss with standard Gaussian noise.*

*Proof.* Let $B = (\beta_1, \ldots, \beta_d) \in \mathbb{R}^{d \times d}$ be the weighted adjacent matrix of a SEM, the linear SEM can be written in the matrix form:

$$X = XB + N \tag{7}$$

where $\mathbb{E}(N|X) = \overrightarrow{\boldsymbol{0}}$, $\mathrm{Var}(N|X) = diag(\sigma_1^2, \ldots, \sigma_d^2)$, and $B_{ii} = 0$ since $X_i$ cannot be the parent of itself. Let $\mathbf{X} \in \mathbb{R}^{n \times d}$ be the observational data and $\mathbf{N} \in \mathbb{R}^{n \times d}$ be the corresponding errors, then

$$\mathbf{X} = \mathbf{X}B + \mathbf{N}.$$

The original and reweighted functions for optimization are

$$S(B; \mathbf{X}) + \mathcal{P}_{DAG}(B) = \mathcal{L}(B; \mathbf{X}) + \lambda \mathcal{R}_{sparse}(B) + \mathcal{P}(B),$$
$$S_{\mathrm{w}}(B; \mathbf{X}) + \mathcal{P}_{DAG}(B) = \mathcal{L}_{\mathrm{w}}(B; \mathbf{X}) + \lambda \mathcal{R}_{sparse}(B) + \mathcal{P}_{DAG}(B).$$

Comparing the above functions, only the first goodness-of-fit term are different, we will only consider this term.

For the least-squares loss case, the optimization problem is

$$\min_{B} \mathcal{L}_{\mathbf{w}}(B; \mathbf{X}) = \min_{B} \sum_{i=1}^{n} w_i l(\mathbf{x}_i, \mathbf{x}_i B),$$

$$s.t. \ B_{ii} = 0, \quad i = 1, \ldots, d.$$

Let $W = diag(w_1, \ldots, w_n)$ be the $n$-dimensional matrix, and rewrite the loss function as

$$
\begin{aligned}
\mathcal{L}_{\mathbf{w}}(B; \mathbf{X}) &= \sum_{i=1}^{n} w_i \|\mathbf{x}_i - \mathbf{x}_i B\|_2^2 \\
&= \sum_{i=1}^{n} w_i (\mathbf{x}_i - \mathbf{x}_i B)(\mathbf{x}_i - \mathbf{x}_i B)^{\top} \\
&= \sum_{i=1}^{n} \sum_{j=1}^{d} w_i (\mathbf{X}_{ij} - \mathbf{x}_i \beta_j)^2 \\
&= \sum_{j=1}^{d} (\mathbf{x}^j - \mathbf{X}\beta_j)^{\top} W (\mathbf{x}^j - \mathbf{X}\beta_j),
\end{aligned}
$$

where $\mathbf{x}^j$ is the $j$-th column in matrix $\mathbf{X}$. Let $D_j$ be the $d$-dimensional identify matrix by setting $j$-th element as 0, for $j = 1, \ldots, d$. The above optimization is able to be written without the restriction:

$$
\begin{aligned}
\min_{B} \tilde{\mathcal{L}}_{\mathbf{w}}(B; \mathbf{X}) &= \min_{B} \sum_{j=1}^{d} (\mathbf{x}^j - \mathbf{X}D_j\beta_j)^{\top} W (\mathbf{x}^j - \mathbf{X}D_j\beta_j) \\
&= \min_{B} \sum_{j=1}^{d} \left( (\mathbf{x}^j)^{\top} W \mathbf{x}^j - 2(\mathbf{x}^j)^{\top} W \mathbf{X}D_j\beta_j + \beta_j^{\top} D_j^{\top} \mathbf{X}^{\top} W \mathbf{X}D_j\beta_j \right).
\end{aligned}
$$

The partial derivative of the loss function with respect to $\beta_j$ is

$$
\begin{aligned}
\frac{\partial \tilde{\mathcal{L}}_{\mathrm{w}}(B;\mathbf{X})}{\partial \beta_j} &= \frac{\partial\left[\sum_{j=1}^d \left((\mathbf{x}^j)^\top W X_j - 2(\mathbf{x}^j)^\top W \mathbf{X} D_j \beta_j + \beta_j^\top D_j^\top \mathbf{X}^\top W \mathbf{X} D_j \beta_j\right)\right]}{\partial \beta_j} \\
&= \frac{\partial\left((\mathbf{x}^j)^\top W \mathbf{x}^j - 2(\mathbf{x}^j)^\top W \mathbf{X} D_j \beta_j + \beta_j^\top D_j^\top \mathbf{X}^\top W \mathbf{X} D_j \beta_j\right)}{\partial \beta_j} \\
&= -2 D_j^\top \mathbf{X}^\top W \mathbf{x}^j + 2 D_j^\top \mathbf{X}^\top W \mathbf{X} D_j \beta_j.
\end{aligned}
$$

Setting the partial derivative to zero produces the optimal parameter:

$$
\begin{aligned}
\hat{\beta}_j &= D_j^\top (\mathbf{X}^\top W \mathbf{X})^{-1} D_j D_j^\top \mathbf{X}^\top W \mathbf{x}^j \\
&= D_j^\top (\mathbf{X}^\top W \mathbf{X})^{-1} D_j D_j^\top \mathbf{X}^\top W (\mathbf{X} D_j \beta_j + \mathbf{N}^j) \\
&= D_j \beta_j + D_j (\mathbf{X}^\top W \mathbf{X})^{-1} \mathbf{X}^\top W \mathbf{N}^j,
\end{aligned}
\tag{8}
$$

where $\mathbf{N}^j \in \mathbb{R}^n$ is the $j$-th column in matrix $\mathbf{N}$. In the above equation, the second equality holds because $\mathbf{x}^j = \mathbf{X} D_j \beta_j + \mathbf{N}^j$. Similarly, one can easily obtain that the optimum parameter for ordinary mean-squared loss is

$$
\tilde{\beta}_j = D_j \beta_j + D_j (\mathbf{X}^\top \mathbf{X})^{-1} \mathbf{X}^\top \mathbf{N}^j.
\tag{9}
$$

It is obvious that the difference between Equation 8 and Equation 9 is the second term. Compute the mean and variance matrix of the second term in Equation 8, we can get

$$
\begin{aligned}
\mathbb{E}\left[(\mathbf{X}^\top W \mathbf{X})^{-1} \mathbf{X}^\top W \mathbf{N}^j\right] &= \mathbb{E}\left(\mathbb{E}\left[(\mathbf{X}^\top W \mathbf{X})^{-1} \mathbf{X}^\top W \mathbf{N}^j \,|\, \mathbf{X}\right]\right) \\
&= \mathbb{E}\left((\mathbf{X}^\top W \mathbf{X})^{-1} \mathbf{X}^\top W \cdot \mathbb{E}\left[\mathbf{N}^j \,|\, \mathbf{X}\right]\right) \\
&= \vec{\mathbf{0}},
\end{aligned}
$$

and

$$
\begin{aligned}
&\mathrm{Var}\left[(\mathbf{X}^\top W \mathbf{X})^{-1} \mathbf{X}^\top W \mathbf{N}^j\right] \\
&= \mathbb{E}\left((\mathbf{X}^\top W \mathbf{X})^{-1} \mathbf{X}^\top W \mathbf{N}^j (\mathbf{N}^j)^\top W^\top \mathbf{X} (\mathbf{X}^\top W \mathbf{X})^{-1}\right) \\
&\qquad - \left(\mathbb{E}\left[(\mathbf{X}^\top W \mathbf{X})^{-1} \mathbf{X}^\top W \mathbf{N}^j\right]\right)\left(\mathbb{E}\left[(\mathbf{X}^\top W \mathbf{X})^{-1} \mathbf{X}^\top W \mathbf{N}^j\right]\right)^\top \\
&= \mathbb{E}\left(\mathbb{E}\left[(\mathbf{X}^\top W \mathbf{X})^{-1} \mathbf{X}^\top W \mathbf{N}^j (\mathbf{N}^j)^\top W \mathbf{X} (\mathbf{X}^\top W \mathbf{X})^{-1} \,|\, \mathbf{X}\right]\right) \\
&= \mathbb{E}\left[(\mathbf{X}^\top W \mathbf{X})^{-1} \mathbf{X}^\top W \cdot \mathbb{E}\left[\mathbf{N}^j (\mathbf{N}^j)^\top \,|\, \mathbf{X}\right] \cdot W \mathbf{X} (\mathbf{X}^\top W \mathbf{X})^{-1}\right] \\
&= \mathbb{E}\left[(\mathbf{X}^\top W \mathbf{X})^{-1} \mathbf{X}^\top W \cdot \mathbb{E}\left[\mathbf{N}^j (\mathbf{N}^j)^\top \,|\, \mathbf{X}\right] \cdot W \mathbf{X} (\mathbf{X}^\top W \mathbf{X})^{-1}\right] \\
&= \sigma_j^2 \mathbb{E}\left[(\mathbf{X}^\top W \mathbf{X})^{-1} \mathbf{X}^\top W^2 \mathbf{X} (\mathbf{X}^\top W \mathbf{X})^{-1}\right].
\end{aligned}
$$

The last equality holds because $\mathbb{E}(NN^\top | X) = \mathrm{Var}(N|X) + \mathbb{E}(N|X)[\mathbb{E}(N|X)]^\top = diag(\sigma_1^2, \ldots, \sigma_d^2)$.

Since $\mathbf{w} \in \mathbb{C}(\tau)$, it is easy to know that the variance matrix is finite. By the Kolmogorov's strong law of large numbers, the second term converges to zero, thus

$$
\hat{\beta}_j \xrightarrow{a.s.} D_j \beta_j,
$$

which is same as the ordinary case. Since noise $N = (N_1, \ldots, N_d)$ are jointly independent, the previous process can be apply to the other $j \in \{1, \ldots, d\}$. Let $\hat{B} = (\hat{\beta}_1, \ldots, \hat{\beta}_d)$ and $\tilde{B} = (\tilde{\beta}_1, \ldots, \tilde{\beta}_d)$, then

$$\hat{B} - \tilde{B} \xrightarrow{a.s.} \mathbf{0}.$$

Therefore, the convergence has been shown for 'case a.'

Since the noise follows a Gaussian distribution, i.e.

$$X - XB = N = (N_1, \ldots, N_d) \sim \mathcal{N}\big(\vec{\mathbf{0}}, diag(\sigma_1^2, \ldots, \sigma_d^2)\big),$$

the loss function (negative log-likelihood function) is

$$
\begin{aligned}
\mathcal{L}_{\mathrm{w}}(B; \mathbf{X}) &= -\sum_{i=1}^{n} w_i \sum_{j=1}^{d} \left[ \log\left(\frac{1}{\sigma_j \sqrt{2\pi}}\right) - \frac{(\mathbf{X}_{ij} - \mathbf{x}_i \beta_j)^2}{2\sigma_j^2} \right] \\
&= \sum_{j=1}^{d} \sum_{i=1}^{n} w_i \log\left(\sigma_j \sqrt{2\pi}\right) + \sum_{j=1}^{d} \sum_{i=1}^{n} \frac{w_i}{2\sigma_j^2} (\mathbf{X}_{ij} - \mathbf{x}_i \beta_j)^2 \\
&= \sum_{j=1}^{d} \sum_{i=1}^{n} w_i \log\left(\sigma_j \sqrt{2\pi}\right) + \sum_{j=1}^{d} \frac{1}{2\sigma_j^2} (\mathbf{x}^j - \mathbf{X}\beta_j)^\top W (\mathbf{x}^j - \mathbf{X}\beta_j). \quad (10)
\end{aligned}
$$

To minimize the loss function above w.r.t. $B$, it is equivalent to minimize the second term in Equation 10:

$$\min_B \mathcal{L}_{\mathrm{w}}(B; \mathbf{X}) \quad \Longleftrightarrow \quad \min_B \sum_{j=1}^{d} \frac{1}{2\sigma_j^2} (\mathbf{x}^j - \mathbf{X}\beta_j)^\top W (\mathbf{x}^j - \mathbf{X}\beta_j).$$

It can be seen that the RHS above is similar to the loss function in 'case a.' except the coefficients $\frac{1}{2\sigma_j^2}, j = 1, \ldots, d$. Therefore, one can use same approaches to get the equivalence result for 'case b.'

Consequently, the proofs of the two special cases have been done. $\qquad \square$

## C.2 PROOF OF THEOREM 2

**Theorem 2.** *Suppose that in the optimization phase, the $i$-th observation has a larger error than the $j$-th observation in the sense that $l(\boldsymbol{x}_i, f(\boldsymbol{x}_i)) > l(\boldsymbol{x}_j, f(\boldsymbol{x}_j))$, where $i, j \in \{1, \ldots, n\}$. Then,*

$$w_i^* \geq w_j^*,$$

*where $w_i^*, w_i^*$ are the optimal weights in Equation 6. The equality holds if and only if $w_i^* = w_j^* = \frac{\tau}{n}$ or $w_i^* = w_j^* = \frac{1}{\tau n}$.*

*Proof.* We will show the theorem by contradiction. Without loss of generality, let $i = 1, j = 2$, and suppose $w_1^* < w_2^*$. Since $\mathrm{w}^* \in \mathbb{C}(\tau)$, one can find a small constant $\varepsilon \in \big(0, min\{w_1^* - \frac{\tau}{n}, \frac{1}{\tau n} - w_2^*\}\big)$, such that

$$\mathrm{w}^{**} = (w_1^* + \varepsilon, w_2^* - \varepsilon, w_3^* \ldots, w_n^*) \in \mathbb{C}(\tau). \quad (11)$$

Therefore,

$$
\begin{aligned}
& S_{\mathrm{w}^*}(\mathcal{G}; \mathbf{X}) - S_{\mathrm{w}^{**}}(\mathcal{G}; \mathbf{X}) \\
&= \big[ w_1^* \cdot l(\mathbf{x}_1, f(\mathbf{x}_1)) + w_2^* \cdot l(\mathbf{x}_2, f(\mathbf{x}_2)) \big] - \big[ (w_1^* + \varepsilon) \cdot l(\mathbf{x}_1, f(\mathbf{x}_1)) + (w_2^* - \varepsilon) \cdot l(\mathbf{x}_2, f(\mathbf{x}_2)) \big] \\
&= \varepsilon \cdot \big[ l(\mathbf{x}_2, f(\mathbf{x}_2)) - l(\mathbf{x}_1, f(\mathbf{x}_1)) \big] < 0,
\end{aligned}
$$

which contradicts $\mathbf{w}^* \in \arg\max_{\mathrm{w}} S_{\mathrm{w}}(\mathcal{G}; \mathbf{X})$. Thus, by contradiction, we can get $w_1^* \geq w_2^*$ as stated in the theorem.

When $\frac{\tau}{n} < w_1^* = w_2^* < \frac{1}{\tau n}$, we can also find a small $\varepsilon \in \big(0, min\{w_1^* - \frac{\tau}{n}, \frac{1}{\tau n} - w_2^*\}\big)$ such that Equation 11 holds. Similarly, we can get $S_{\mathrm{w}^*}(\mathcal{G}; \mathbf{X}) < S_{\mathrm{w}^{**}}(\mathcal{G}; \mathbf{X})$, and $w_1^* = w_2^* = \frac{\tau}{n}$ or $w_1^* = w_2^* = \frac{1}{\tau n}$ by contradiction. $\qquad \square$

# D  SUPPLEMENTARY EXPERIMENTS

## D.1  BASELINES

We select seven state-of-the-art causal discovery methods as baselines for comparison:

- **NOTEARS** (Zheng et al., 2018) is a breakthrough work that firstly recasts the combinatorial graph search problem as a continuous optimization problem in linear settings. NOTEARS estimates the true causal graph by minimizing the reconstruction loss with the continuous acyclicity constraint.

- **NOTEARS-MLP** (Zheng et al., 2020) is an extension of NOTEARS for nonlinear settings, approximating the generative SEM model by MLP while only applying the continuous acyclicity constraint to the first layer of the MLP.

- **GraN-DAG** (Lachapelle et al., 2020) adapts the continuous constrained optimization formulation to allow for nonlinear relationships between variables using neural networks and makes use of a final pruning step to remove spurious edges, thus achieving good results in nonlinear settings.

- **GOLEM** (Ng et al., 2020) improves on the least squares score function (Zheng et al., 2018) by proposing a score function that directly maximizes the data likelihood. They show the likelihood-based score function with soft sparsity regularization is sufficient to asymptotically learn a DAG equivalent to the ground-truth DAG.

- **DICD** (Wang et al., 2022) aims to discover the environment-invariant causation while removing the environment-dependent correlation based on ground truth domain annotation.

- **CD-NOD** (Huang et al., 2020) is a constrained-based causal discovery method that is designed for heterogeneous data, *i.e.,* datasets from different environments. CD-NOD utilizes the independent changes across environments to predict the causal orientations and proposes constrained-based and kernel-based methods to find the causal structure.

- **GES** (Chickering, 2002) is a score-based search algorithm that searches over the space of equivalence classes of Bayesian network structures.

## D.2  EXPERIMENTAL SETTINGS

For NOTEARS, we follow the original linear implementation. For GOLEM, we adopt the GOLEM-NV setting from the original repo. For NOTEARS-MLP, we follow the original non-linear implementation which consist a Multilayer Perceptron (MLP) comprising of two hidden layers with ten neurons each and ReLU activation functions (except for the Sachs dataset, which uses only one hidden layer, inherent the settings from Zheng et al. (2020)). For GraN-DAG, we employ the pns, training, and cam-pruning stages from the original code and tune three pipeline stages together for best performance. The ReScore adaptive weights learning model for all nonlinear baselines consists of two hidden layer and ReLU activation, and for linear baselines the layer size is reduced to one. All Experiments are conducted on a single Tesla V100 GPU. Detailed hyperparameter search space for different methods is shown in Table 4.

## D.3  STUDY ON RESCORE

### D.3.1  ILLUSTRATIVE EXAMPLES OF RESCORE

**Motivations.** To fully comprehend the benefits of reweighting, two research hypotheses need to be verified. First, we have to determine the validity of the fundamental understanding of ReScore, which states that real-world datasets inevitably include samples of varying importance. In other words, there are many informative samples that come from disadvantaged groups in real-world scenarios. Additionally, we must confirm that the adaptive weights learned by ReScore are the faithful reflection of sample importance, *i.e.,* less-fitted samples typically come from disadvantaged groups, which are more important than those well-fitted samples.

**Simulations.** Real-world Sachs (Sachs et al., 2005) dataset naturally contains nine groups, where each group corresponds with a different experimental condition. We first rank the importance of

Table 4: Hyperparameter search spaces for each algorithm.

| | **Hyperparameter space** |
|---|---|
| **NOTEARS / NOTEARS+ReScore** | $\lambda \sim \{0.002, 0.005, 0.01, 0.015, 0.02, 0.03, 0.09, 0.1, 0.25\}$
Gumbel softmax temperature $\sim \{0.1, 1, 5, 10, 20, 30, 40, 50, 100\}$
Cut-off threshold $\tau \sim \{0.01, 0.1, 0.3, 0.5, 0.7, 0.9, 0.99\}$
Constraint convergence tolerance $\sim \{10^{-6}, 10^{-8}, 10^{-10}\}$
Log(learning rate of ReScore) $\sim U[-1, -5]$ |
| **GOLEM / GOLEM+ReScore** | $\lambda \sim \{0.002, 0.005, 0.01, 0.015, 0.02, 0.03, 0.09, 0.1, 0.25\}$
Gumbel softmax temperature $\sim \{0.1, 1, 5, 10, 20, 30, 40, 50, 100\}$
Cut-off threshold $\tau \sim \{0.01, 0.1, 0.3, 0.5, 0.7, 0.9, 0.99\}$
Log(learning rate of ReScore) $\sim U[-1, -5]$ |
| **NOTEARS-MLP / NOTEARS-MLP+ReScore** | $\lambda \sim \{0.002, 0.005, 0.01, 0.015, 0.02, 0.03, 0.09, 0.1, 0.25\}$
Gumbel softmax temperature $\sim \{0.1, 1, 5, 10, 20, 30, 40, 50, 100\}$
# hidden units of ReScore $\sim \{1, 10, 20, 50, 80, 100\}$
# hidden layers of ReScore $\sim \{1, 2, 3, 4\}$
# hidden units of NOTEARS-MLP $\sim \{1, 10, 20, 50, 80, 100\}$
# hidden layers of ReScore $\sim \{1, 2, 3\}$
Cut-off threshold $\tau \sim \{0.01, 0.1, 0.3, 0.5, 0.7, 0.9, 0.99\}$
Constraint convergence tolerance $\sim \{10^{-6}, 10^{-8}, 10^{-10}\}$
Log(learning rate of ReScore) $\sim U[-1, -5]$ |
| **GraN-DAG / GraN-DAG+ReScore** | $\lambda \sim \{0.002, 0.005, 0.01, 0.015, 0.02, 0.03, 0.09, 0.1, 0.25\}$
Gumbel softmax temperature $\sim \{0.1, 1, 5, 10, 20, 30, 40, 50, 100\}$
# hidden units of ReScore $\sim \{1, 10, 20, 50, 80, 100\}$
# hidden layers of ReScore $\sim \{1, 2, 3, 4\}$
Log(learning rate of ReScore) $\sim U[-1, -5]$
PNS threshold $\sim \{0.5, 0.75, 1, 2\}$
Log(Pruning cutoff) $\sim \{0.001, 0.005, 0.01, 0.03, 0.1, 0.2, 0.3\}$ |
| **GES / GES+ReScore** | $\lambda \sim \{0.002, 0.005, 0.01, 0.015, 0.02, 0.03, 0.09, 0.1, 0.25\}$
Gumbel softmax temperature $\sim \{0.1, 1, 5, 10, 20, 30, 40, 50, 100\}$
# hidden units of ReScore $\sim \{1, 10, 20, 50, 80, 100\}$
# hidden layers of ReScore $\sim \{1, 2, 3, 4\}$
Cut-off threshold $\tau \sim \{0.01, 0.1, 0.3, 0.5, 0.7, 0.9, 0.99\}$ |

Table 5: Performance comparison for removing samples in different groups

| Group Index | 3 | 5 | 7 | 1 | 2 | 6 | 4 | 8 | 0 |
|---|---|---|---|---|---|---|---|---|---|
| Avg. ranking | 578.4 | 2856.7 | 3368.1 | 3877.0 | 3949.4 | 4549.4 | 4573.2 | 4590.6 | 4910.1 |
| SHD w/o group | 16 | 16 | 17 | 16 | 16 | 17 | 17 | 19 | 19 |
| TPR w/o group | 0.529 | 0.412 | 0.412 | 0.412 | 0.412 | 0.412 | 0.412 | 0.353 | 0.294 |

each group in Sachs by using the average weights for each group learned by ReScore as the criterion. Then we eliminate 500 randomly selected samples in one specific group, perform NOTEARS-MLP, and show its DAG accuracy inferred from the remaining samples. Note that the sample size in each group, which ranges from 700 to 900, is fairly balanced.

**Results.** Table 5 clearly shows a declining trend *w.r.t.* SHD and TPR metrics as the significance of deleting groups grows. Specifically, removing samples from disadvantaged groups such as Groups 8 and 0, which have the highest average weights, will significantly influence the DAG learning quality. In contrast, the SHD and TPR of NOTEARS-MLP can even be maintained or slightly decreased by excluding the samples from groups with relatively low average weights. This illustrates that samples of different importance are naturally present in real-world datasets, and ReScore is capable of successfully extracting this importance.

Table 6: SHD for $p_{corrupt}$ percentage noise samples.

| | 0 | 0.01 | 0.02 | 0.05 | 0.08 | 0.1 | 0.2 | 0.3 | 0.5 |
|---|---|---|---|---|---|---|---|---|---|
| NOTEARS-MLP | 14.9 | 15.2 | 15.3 | 18.9 | 19.8 | 21.3 | 23.9 | **23.8** | **28.3** |
| + ReScore ($\tau \to 0$) | 13.8 | 14.2 | 15.0 | 18.3 | 19.5 | 20.7 | 24.0 | 24.4 | 29.3 |
| + ReScore (Optimal $\tau$) | **13.7** | **14.1** | **15.0** | **18.1** | **19.2** | **19.9** | **21.9** | 24.0 | 28.9 |
| Imp. % | +8% | +7% | +2% | +4% | +3% | +7% | +8% | -1% | -2% |

### D.3.2 SENSITIVITY TO PURE NOISE SAMPLES

**Motivations.** A basic assumption of ReScore is that no pure noise outliers are involved in the training process. Otherwise, the DAG learner might get overwhelmed by arbitrarily up-weighting less well-fitted samples, in this case, pure noise data. The good news is that the constraint of the cutoff threshold $\tau \in \mathbb{C}(\tau) = \{\mathbf{w} : 0 < \frac{\tau}{n} \le w_1, \ldots, w_n \le \frac{1}{\tau n}, \sum_{i=1}^{n} w_i = 1\}$ prevents over-exploitation of pure noise samples, which further strengthens ReScore's ability to withstand outliers. To evaluate the robustness of ReScore against pure noise samples, the following experiments are conducted.

**Simulations.** We produce $p_{corrupt}$ percentage pure noise samples in nonlinear settings ($n = 2000$, $d = 20$, ER2), where those noise samples are generated from a different structural causal model. We try out a broad range of $p_{corrupt} = \{0, 0.01, 0.02, 0.05, 0.08, 0.1, 0.2, 0.3, 0.5\}$.

**Results.** Table 6 reports the comparison of performance in NOTEARS-MLP and two ReScore methods (no cut-off threshold and optimal $\tau$) when encountering pure noise data. The best-performing methods are bold; Imp.% measures the relative improvements of ReScore (Optimal $\tau$) over the backbone NOTEARS-MLP. We observe that ReScore (Optimal $\tau$) consistently yields remarkable improvements compared with NOTEARS-MLP in the case that less than 20% of samples are corrupted. These results demonstrate the robustness of ReScore when handling data that contains a small proportion of pure noise data. Surprisingly, when the cutoff threshold $\tau$ is set to be close to 0, the ReScore can still achieve relative gains over the baseline when less than 10% of the samples are pure noise. Although it is more sensitive to noise samples than the optimum cutoff threshold $\tau$. These surprising findings support the effectiveness of adaptive weights and show the potential of ReScore.

### D.3.3 EFFECT OF HYPERPARAMETER $\tau$.

We investigate the effect of cut-off threshold $\tau$ on the performance of ReScore. Intuitively, ReScore relies on the hyperparameter $\tau$ to control the balance between hard sample mining and robustness towards extremely noisy samples. On one hand, setting the threshold closer to 0 results in no weight-clipping and leaves the model susceptible to noises, which results in sub-optimal performance. On the other hand, setting the threshold closer to 1 disables the reweighting scheme and eventually reduces ReScore performance to its backbone model.

We conduct experiments under different settings of $\tau$ using $n = 2000$ samples generated from GP model on ER4 graphs with $d = 20$ nodes. The weight distribution under best performing threshold $tau = 0.9$ and the trend of SHD *w.r.t.* to $\tau$ is shown in Figure 4. One can observe that ReScore obtains its best performance at $\tau = 0.9$, while a smaller or bigger threshold results in sub-optimal performance. Furthermore, we find that in different settings, the optimal threshold $\tau$ usually falls in the range of $[0.7, 0.99]$. This indicates that ReScore performs best when adaptive reweighting is conducted within a restricted range.

### D.3.4 SENSITIVITY TO NEURAL NETWORK COMPLEXITY.

We also investigated the effect of number of hidden units in our adaptive weights learning model for ReScore. We plot the TPR, FDR, SHD, SID with varing number of hidden units ranging from 10 to 100 units in nonlinear settings, using $n = 600$ and $n = 2,000$ samples generated from GP model on ER4 graph with $d = 10$ nodes. Detailed results could be found in Figure 5. One can first observe our model is stable when increasing the neurons, illustrating the insensitivity of ReScore *w.r.t.* the number of neurons in adaptive weights learning model. On the other hand, more observational

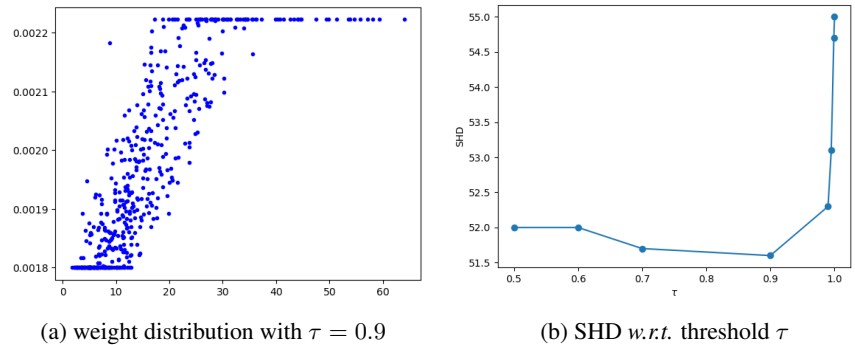

(a) weight distribution with $\tau = 0.9$      (b) SHD *w.r.t.* threshold $\tau$

Figure 4: Study of varying $\tau$ in ReScore model.

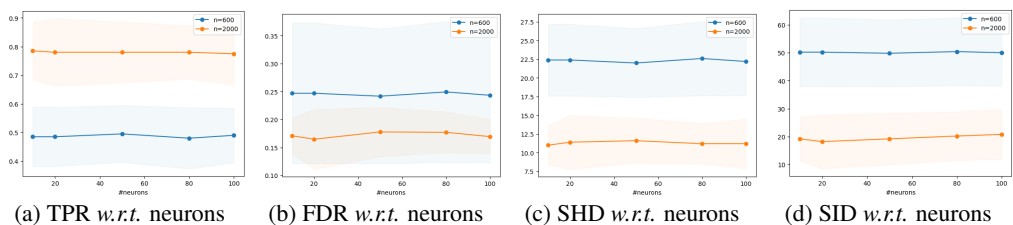

(a) TPR *w.r.t.* neurons    (b) FDR *w.r.t.* neurons    (c) SHD *w.r.t.* neurons    (d) SID *w.r.t.* neurons

Figure 5: Performance with varying neurons in ReScore model.

samples to estimate the parameters could help the ReScore achieve higher performance, indicating rich samples bring benefit.

### D.3.5 TRAINING COSTS.

In terms of time complexity, as shown in Table 7, we report the time for each baseline and ReScore on Sachs. Compared with backbone methods, ReScore adds very little computing cost to training.

### D.4 MORE EXPERIMENTAL RESULTS FOR RQ1

**Discussions.** More experimental results on both the linear and nonlinear synthetic data are reported in Figures 6 - 8 and Tables 8 - 11. The error bars depict the standard deviation across datasets over ten trails. The red and blue percentages separately refer to the increase and decrease of ReScore relative to the original score-based causal discovery methods in each metric. The best performing methods per task are bold.

Table 7: Training cost on Sachs (seconds per iteration/in total).

| NOTEARS | 0.74 / 2.97 |
|---|---|
| + ReScore | 3.8 / 15.3 |
| NOTEARS-MLP | 0.87 / 3.48 |
| + ReScore | 4.3 / 17.0 |
| GOLEM | 13.4 / 53.5 |
| + ReScore | 14.6 / 58.2 |
| GraN-DAG | 4.9 / 197.3 |
| + ReScore | 5.5 / 221.6 |

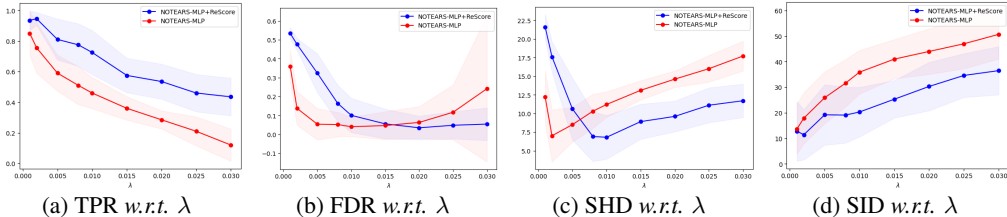

(a) TPR *w.r.t.* $\lambda$     (b) FDR *w.r.t.* $\lambda$     (c) SHD *w.r.t.* $\lambda$     (d) SID *w.r.t.* $\lambda$

Figure 6: Performance comparison between NOTEARS-MLP and ReScore on ER2 graphs of 10 nodes on nonlinear synthetic datasets. The hyperparameter $\lambda$ defined in Equation 2 refers to the graph sparsity.

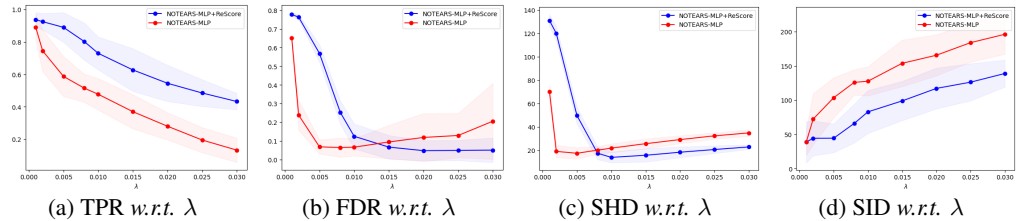

(a) TPR *w.r.t.* $\lambda$     (b) FDR *w.r.t.* $\lambda$     (c) SHD *w.r.t.* $\lambda$     (d) SID *w.r.t.* $\lambda$

Figure 7: Performance comparison between NOTEARS-MLP and ReScore on ER2 graphs of 20 nodes on nonlinear synthetic datasets. The hyperparameter $\lambda$ defined in Equation 2 refers to the graph sparsity.

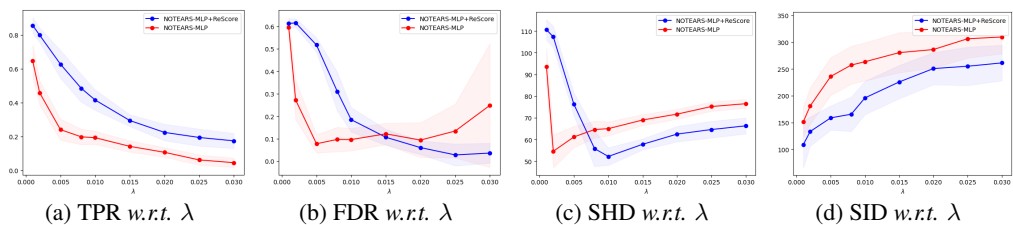

(a) TPR *w.r.t.* $\lambda$     (b) FDR *w.r.t.* $\lambda$     (c) SHD *w.r.t.* $\lambda$     (d) SID *w.r.t.* $\lambda$

Figure 8: Performance comparison between NOTEARS-MLP and ReScore on ER4 graphs of 20 nodes on nonlinear synthetic datasets. The hyperparameter $\lambda$ defined in Equation 2 refers to the graph sparsity.

Table 8: Results for ER graphs of 20 nodes on linear and nonlinear synthetic datasets.

| | ER2 | | | | ER4 | | | |
|---|---|---|---|---|---|---|---|---|
| | TPR ↑ | FDR ↓ | SHD ↓ | SID ↓ | TPR ↑ | FDR ↓ | SHD ↓ | SID ↓ |
| **Random** | $0.11_{\pm0.09}$ | $0.89_{\pm0.08}$ | $56.8_{\pm8.7}$ | $292.3_{\pm45.7}$ | $0.07_{\pm0.03}$ | $0.90_{\pm0.08}$ | $86.9_{\pm7.0}$ | $387.5_{\pm52.3}$ |
| **NOTEARS** | $0.85_{\pm0.08}$ | $\mathbf{0.09}_{\pm0.03}$ | $9.2_{\pm3.8}$ | $55.4_{\pm31.1}$ | $0.74_{\pm0.02}$ | $\mathbf{0.23}_{\pm0.03}$ | $39.4_{\pm7.9}$ | $185.8_{\pm38.1}$ |
| **+ ReScore** | $\mathbf{0.87}_{\pm0.07}^{+2\%}$ | $0.11_{\pm0.05}^{-17\%}$ | $\mathbf{8.8}_{\pm3.5}^{+5\%}$ | $\mathbf{50.6}_{\pm26.3}^{+9\%}$ | $\mathbf{0.79}_{\pm0.05}^{+7\%}$ | $0.28_{\pm0.05}^{-17\%}$ | $\mathbf{36.8}_{\pm7.9}^{+7\%}$ | $\mathbf{180.8}_{\pm43.5}^{+3\%}$ |
| **GOLEM** | $0.75_{\pm0.07}$ | $0.20_{\pm0.11}$ | $17.0_{\pm6.1}$ | $78.2_{\pm22.6}$ | $0.46_{\pm0.06}$ | $0.50_{\pm0.06}$ | $73.6_{\pm7.9}$ | $249.8_{\pm7.8}$ |
| **+ ReScore** | $0.76_{\pm0.06}^{+2\%}$ | $0.20_{\pm0.10}^{+1\%}$ | $15.8_{\pm5.8}^{+8\%}$ | $77.0_{\pm21.5}^{+2\%}$ | $0.48_{\pm0.06}^{+3\%}$ | $0.43_{\pm0.06}^{+16\%}$ | $70.2_{\pm8.3}^{+5\%}$ | $246.2_{\pm11.4}^{+1\%}$ |
| **NOTEARS-MLP** | $0.70_{\pm0.12}$ | $0.13_{\pm0.07}$ | $14.9_{\pm5.4}$ | $98.4_{\pm22.5}$ | $\mathbf{0.44}_{\pm0.09}$ | $0.26_{\pm0.10}$ | $55.0_{\pm9.2}$ | $176.3_{\pm33.3}$ |
| **+ ReScore** | $0.73_{\pm0.09}^{+3\%}$ | $0.11_{\pm0.05}^{+7\%}$ | $13.7_{\pm5.1}^{+8\%}$ | $88.8_{\pm23.3}^{+11\%}$ | $0.41_{\pm0.07}^{-6\%}$ | $0.17_{\pm0.08}^{+54\%}$ | $\mathbf{51.6}_{\pm6.4}^{+7\%}$ | $179.9_{\pm33.7}^{-2\%}$ |
| **GraN-DAG** | $\mathbf{0.81}_{\pm0.15}$ | $0.08_{\pm0.08}$ | $9.3_{\pm5.4}$ | $53.4_{\pm24.4}$ | $0.20_{\pm0.07}$ | $0.18_{\pm0.08}$ | $57.4_{\pm4.6}$ | $131.5_{\pm21.4}$ |
| **+ ReScore** | $\mathbf{0.81}_{\pm0.14}^{+0\%}$ | $\mathbf{0.05}_{\pm0.04}^{+64\%}$ | $\mathbf{8.5}_{\pm5.7}^{+9\%}$ | $\mathbf{51.0}_{\pm24.6}^{+5\%}$ | $0.21_{\pm0.07}^{+5\%}$ | $\mathbf{0.17}_{\pm0.09}^{+8\%}$ | $56.2_{\pm4.6}^{+2\%}$ | $\mathbf{125.4}_{\pm23.3}^{+5\%}$ |

Table 9: Results for ER graphs of 50 nodes on linear and nonlinear synthetic datasets.

| | ER2 | | | | ER4 | | | |
|---|---|---|---|---|---|---|---|---|
| | TPR ↑ | FDR ↓ | SHD ↓ | SID ↓ | TPR ↑ | FDR ↓ | SHD ↓ | SID ↓ |
| **Random** | $0.04_{\pm0.02}$ | $0.90_{\pm0.03}$ | $397.3_{\pm12.7}$ | $1082.0_{\pm182.2}$ | $0.09_{\pm0.08}$ | $0.92_{\pm0.08}$ | $998.2_{\pm45.9}$ | $3399.1_{\pm489.2}$ |
| **NOTEARS** | $0.79_{\pm0.06}$ | $0.09_{\pm0.03}$ | $27.6_{\pm7.7}$ | $427.0_{\pm186.1}$ | $0.51_{\pm0.12}$ | $0.27_{\pm0.10}$ | $133.4_{\pm29.5}$ | $1643.8_{\pm172.2}$ |
| **+ ReScore** | $\mathbf{0.88}_{\pm0.06}^{+11\%}$ | $0.15_{\pm0.04}^{-39\%}$ | $\mathbf{26.2}_{\pm7.6}^{+5\%}$ | $\mathbf{266.0}_{\pm146.4}^{+61\%}$ | $\mathbf{0.52}_{\pm0.21}^{+3\%}$ | $0.29_{\pm0.07}^{-7\%}$ | $\mathbf{130.2}_{\pm37.4}^{+2\%}$ | $\mathbf{1453.6}_{\pm336.5}^{+13\%}$ |
| **GOLEM** | $0.80_{\pm0.09}$ | $0.35_{\pm0.09}$ | $68.6_{\pm19.7}$ | $433.5_{\pm215.6}$ | $0.31_{\pm0.11}$ | $0.68_{\pm0.06}$ | $150.6_{\pm25.1}$ | $1775.4_{\pm161.6}$ |
| **+ ReScore** | $0.82_{\pm0.15}^{+3\%}$ | $0.33_{\pm0.14}^{+5\%}$ | $63.4_{\pm27.9}^{+8\%}$ | $430.2_{\pm155.5}^{+1\%}$ | $0.39_{\pm0.06}^{+24\%}$ | $0.66_{\pm0.06}^{+3\%}$ | $146.3_{\pm26.3}^{+3\%}$ | $1643.6_{\pm114.8}^{+8\%}$ |
| **NOTEARS-MLP** | $0.32_{\pm0.04}$ | $0.13_{\pm0.08}$ | $69.5_{\pm4.7}$ | $884.4_{\pm172.8}$ | $0.17_{\pm0.02}$ | $\mathbf{0.06}_{\pm0.04}$ | $167.0_{\pm4.1}$ | $1607.6_{\pm97.0}$ |
| **+ ReScore** | $0.51_{\pm0.08}^{+59\%}$ | $\mathbf{0.10}_{\pm0.07}^{+30\%}$ | $53.5_{\pm8.7}^{+30\%}$ | $628.1_{\pm120.6}^{+41\%}$ | $0.26_{\pm0.04}^{+52\%}$ | $0.11_{\pm0.05}^{-51\%}$ | $154.4_{\pm6.4}^{+8\%}$ | $1437.7_{\pm111.1}^{+12\%}$ |
| **GraN-DAG** | $0.52_{\pm0.09}$ | $0.15_{\pm0.05}$ | $51.6_{\pm9.3}$ | $632.8_{\pm140.3}$ | $\mathbf{0.32}_{\pm0.04}$ | $0.08_{\pm0.16}$ | $141.6_{\pm8.2}$ | $1379.0_{\pm91.3}$ |
| **+ ReScore** | $\mathbf{0.53}_{\pm0.06}^{+3\%}$ | $0.11_{\pm0.02}^{+36\%}$ | $\mathbf{46.0}_{\pm6.0}^{+12\%}$ | $\mathbf{581.0}_{\pm104.7}^{+9\%}$ | $0.31_{\pm0.03}^{-4\%}$ | $0.06_{\pm0.04}^{+32\%}$ | $\mathbf{138.8}_{\pm7.5}^{+2\%}$ | $\mathbf{1351.0}_{\pm98.2}^{+2\%}$ |

Table 10: Results for SF graphs of 10 nodes on linear and nonlinear synthetic datasets.

| | SF2 | | | | SF4 | | | |
|---|---|---|---|---|---|---|---|---|
| | TPR ↑ | FDR ↓ | SHD ↓ | SID ↓ | TPR ↑ | FDR ↓ | SHD ↓ | SID ↓ |
| **Random** | $0.05_{\pm0.03}$ | $0.91_{\pm0.09}$ | $32.2_{\pm7.97}$ | $35.1_{\pm7.3}$ | $0.13_{\pm0.01}$ | $0.93_{\pm0.15}$ | $57.2_{\pm10.3}$ | $79.1_{\pm8.7}$ |
| **NOTEARS** | $0.98_{\pm0.02}$ | $\mathbf{0.02}_{\pm0.03}$ | $0.8_{\pm0.5}$ | $\mathbf{1.0}_{\pm2.0}$ | $0.95_{\pm0.03}$ | $0.03_{\pm0.02}$ | $12.2_{\pm1.2}$ | $6.2_{\pm5.3}$ |
| **+ ReScore** | $\mathbf{0.99}_{\pm0.02}^{+1\%}$ | $0.04_{\pm0.04}^{-45\%}$ | $\mathbf{0.4}_{\pm0.7}^{+100\%}$ | $\mathbf{1.0}_{\pm0.9}^{+0\%}$ | $\mathbf{0.97}_{\pm0.03}^{+2\%}$ | $0.03_{\pm0.03}^{+27\%}$ | $10.2_{\pm1.5}^{+20\%}$ | $\mathbf{3.0}_{\pm1.9}^{+107\%}$ |
| **GOLEM** | $0.96_{\pm0.07}$ | $0.07_{\pm0.12}$ | $1.8_{\pm3.1}$ | $1.2_{\pm2.4}$ | $0.85_{\pm0.03}$ | $0.12_{\pm0.08}$ | $7.0_{\pm2.3}$ | $12.8_{\pm7.9}$ |
| **+ ReScore** | $0.97_{\pm0.07}^{+1\%}$ | $0.07_{\pm0.12}^{+3\%}$ | $1.4_{\pm2.9}^{+29\%}$ | $1.2_{\pm2.4}^{+0\%}$ | $0.87_{\pm0.06}^{+3\%}$ | $\mathbf{0.10}_{\pm0.08}^{+17\%}$ | $\mathbf{5.8}_{\pm2.9}^{+21\%}$ | $9.8_{\pm8.2}^{+31\%}$ |
| **NOTEARS-MLP** | $\mathbf{0.84}_{\pm0.17}$ | $0.25_{\pm0.12}$ | $6.7_{\pm3.4}$ | $8.1_{\pm7.3}$ | $0.73_{\pm0.14}$ | $0.23_{\pm0.05}$ | $12.0_{\pm3.9}$ | $19.4_{\pm7.4}$ |
| **+ ReScore** | $0.82_{\pm0.22}^{-2\%}$ | $0.17_{\pm0.08}^{+45\%}$ | $5.8_{\pm3.3}^{+16\%}$ | $\mathbf{6.0}_{\pm3.8}^{+35\%}$ | $\mathbf{0.88}_{\pm0.03}^{+20\%}$ | $0.27_{\pm0.07}^{-16\%}$ | $12.8_{\pm9.3}^{+9\%}$ | $12.8_{\pm9.3}^{+52\%}$ |
| **GraN-DAG** | $0.69_{\pm0.20}$ | $0.05_{\pm0.05}$ | $5.9_{\pm3.0}$ | $12.0_{\pm8.2}$ | $0.82_{\pm0.11}$ | $\mathbf{0.11}_{\pm0.08}$ | $8.7_{\pm1.8}$ | $8.4_{\pm4.1}$ |
| **+ ReScore** | $0.72_{\pm0.17}^{+4\%}$ | $\mathbf{0.04}_{\pm0.03}^{+28\%}$ | $\mathbf{5.3}_{\pm2.8}^{+11\%}$ | $10.5_{\pm8.7}^{+14\%}$ | $0.86_{\pm0.12}^{+5\%}$ | $0.12_{\pm0.08}^{-12\%}$ | $\mathbf{8.1}_{\pm2.0}^{+7\%}$ | $\mathbf{7.0}_{\pm6.7}^{+20\%}$ |

Table 11: Results for SF graphs of 20 nodes on linear and nonlinear synthetic datasets.

| | SF2 | | | | SF4 | | | |
|---|---|---|---|---|---|---|---|---|
| | TPR ↑ | FDR ↓ | SHD ↓ | SID ↓ | TPR ↑ | FDR ↓ | SHD ↓ | SID ↓ |
| **Random** | $0.11_{\pm0.10}$ | $0.89_{\pm0.03}$ | $43.2_{\pm5.4}$ | $96.8_{\pm10.4}$ | $0.09_{\pm0.05}$ | $0.88_{\pm0.05}$ | $108.2_{\pm12.9}$ | $155.6_{\pm37.2}$ |
| **NOTEARS** | $0.90_{\pm0.06}$ | $\mathbf{0.02}_{\pm0.01}$ | $4.0_{\pm1.9}$ | $19.8_{\pm12.8}$ | $0.90_{\pm0.05}$ | $0.12_{\pm0.06}$ | $45.2_{\pm7.0}$ | $28.6_{\pm20.2}$ |
| **+ ReScore** | $0.95_{\pm0.04}^{+6\%}$ | $0.06_{\pm0.04}^{-70\%}$ | $\mathbf{3.6}_{\pm1.8}^{+11\%}$ | $\mathbf{9.8}_{\pm8.1}^{+102\%}$ | $\mathbf{0.93}_{\pm0.03}^{+3\%}$ | $\mathbf{0.02}_{\pm0.07}^{+624\%}$ | $45.0_{\pm6.8}^{+0\%}$ | $\mathbf{25.6}_{\pm12.1}^{+12\%}$ |
| **GOLEM** | $0.96_{\pm0.03}$ | $0.19_{\pm0.06}$ | $9.0_{\pm3.2}$ | $10.4_{\pm7.0}$ | $0.83_{\pm0.05}$ | $0.35_{\pm0.09}$ | $42.8_{\pm13.0}$ | $41.4_{\pm14.8}$ |
| **+ ReScore** | $\mathbf{0.96}_{\pm0.02}^{+0\%}$ | $0.18_{\pm0.06}^{+4\%}$ | $8.6_{\pm3.1}^{+5\%}$ | $10.4_{\pm7.0}^{+0\%}$ | $0.85_{\pm0.43}^{+2\%}$ | $0.34_{\pm0.09}^{+5\%}$ | $\mathbf{39.8}_{\pm14.0}^{+8\%}$ | $37.6_{\pm12.8}^{+10\%}$ |
| **NOTEARS-MLP** | $\mathbf{0.42}_{\pm0.13}$ | $0.23_{\pm0.13}$ | $25.5_{\pm4.5}$ | $49.9_{\pm7.4}$ | $0.20_{\pm0.03}$ | $0.22_{\pm0.12}$ | $58.9_{\pm3.1}$ | $115.6_{\pm25.0}$ |
| **+ ReScore** | $0.41_{\pm0.13}^{-2\%}$ | $\mathbf{0.10}_{\pm0.10}^{+121\%}$ | $23.5_{\pm4.5}^{+9\%}$ | $47.6_{\pm9.4}^{+5\%}$ | $\mathbf{0.21}_{\pm0.04}^{+3\%}$ | $\mathbf{0.09}_{\pm0.09}^{+131\%}$ | $\mathbf{56.4}_{\pm2.2}^{+4\%}$ | $\mathbf{109.0}_{\pm21.8}^{+6\%}$ |
| **GraN-DAG** | $0.03_{\pm0.15}$ | $0.24_{\pm0.17}$ | $27.1_{\pm4.15}$ | $77.0_{\pm28.0}$ | $0.20_{\pm0.06}$ | $0.18_{\pm0.12}$ | $56.8_{\pm4.5}$ | $133.4_{\pm21.0}$ |
| **+ ReScore** | $0.03_{\pm0.15}^{-6\%}$ | $0.15_{\pm0.10}^{+63\%}$ | $25.7_{\pm4.4}^{+5\%}$ | $72.8_{\pm26.0}^{+6\%}$ | $0.21_{\pm0.07}^{+2\%}$ | $0.17_{\pm0.08}^{+8\%}$ | $56.4_{\pm4.6}^{+1\%}$ | $125.4_{\pm23.3}^{+6\%}$ |

Table 12: Results for SF graphs of 50 nodes on linear and nonlinear synthetic datasets.

| | SF2 | | | | SF4 | | | |
|---|---|---|---|---|---|---|---|---|
| | TPR ↑ | FDR ↓ | SHD ↓ | SID ↓ | TPR ↑ | FDR ↓ | SHD ↓ | SID ↓ |
| **Random** | $0.10_{\pm0.08}$ | $0.89_{\pm0.07}$ | $334.2_{\pm16.9}$ | $1093.3_{\pm145.4}$ | $0.12_{\pm0.11}$ | $0.89_{\pm0.04}$ | $1023.5_{\pm49.5}$ | $1903.9_{\pm194.3}$ |
| **NOTEARS** | $0.82_{\pm0.03}$ | $\mathbf{0.07}_{\pm0.05}$ | $23.6_{\pm6.2}$ | $135.4_{\pm47.5}$ | $0.71_{\pm0.18}$ | $0.25_{\pm0.07}$ | $97.6_{\pm36.9}$ | $276.2_{\pm131.0}$ |
| **+ ReScore** | $\mathbf{0.94}_{\pm0.03}^{+15\%}$ | $0.15_{\pm0.06}^{-55\%}$ | $21.6_{\pm9.1}^{+9\%}$ | $61.2_{\pm22.6}^{+121\%}$ | $\mathbf{0.73}_{\pm0.05}^{+3\%}$ | $\mathbf{0.10}_{\pm0.03}^{+138\%}$ | $\mathbf{67.6}_{\pm12.3}^{+44\%}$ | $\mathbf{275.2}_{\pm55.0}^{+0\%}$ |
| **GOLEM** | $0.77_{\pm0.07}$ | $0.19_{\pm0.11}$ | $38.6_{\pm16.7}$ | $161.6_{\pm53.2}$ | $0.62_{\pm0.17}$ | $0.21_{\pm0.09}$ | $114.2_{\pm37.5}$ | $384.0_{\pm107.4}$ |
| **+ ReScore** | $0.79_{\pm0.09}^{+2\%}$ | $0.24_{\pm0.12}^{-20\%}$ | $32.2_{\pm11.1}^{+20\%}$ | $143.4_{\pm63.0}^{+13\%}$ | $0.68_{\pm0.17}^{+9\%}$ | $0.21_{\pm0.09}^{+1\%}$ | $113.7_{\pm37.5}^{+0\%}$ | $366.4_{\pm107.0}^{+5\%}$ |
| **NOTEARS-MLP** | $0.22_{\pm0.04}$ | $0.04_{\pm0.04}$ | $75.8_{\pm4.0}$ | $266.8_{\pm46.0}$ | $0.11_{\pm0.02}$ | $\mathbf{0.03}_{\pm0.02}$ | $168.8_{\pm3.8}$ | $461.6_{\pm54.9}$ |
| **+ ReScore** | $\mathbf{0.23}_{\pm0.05}^{+4\%}$ | $0.07_{\pm0.07}^{-47\%}$ | $\mathbf{75.6}_{\pm4.3}^{+0\%}$ | $267.2_{\pm36.6}^{-0\%}$ | $\mathbf{0.13}_{\pm0.05}^{+10\%}$ | $0.07_{\pm0.06}^{-52\%}$ | $\mathbf{167.7}_{\pm7.0}^{+1\%}$ | $\mathbf{453.4}_{\pm57.7}^{+2\%}$ |
| **GraN-DAG** | $0.19_{\pm0.03}$ | $0.28_{\pm0.05}$ | $80.2_{\pm3.5}$ | $380.8_{\pm56.1}$ | $0.11_{\pm0.03}$ | $0.25_{\pm0.11}$ | $171.4_{\pm6.3}$ | $549.6_{\pm84.9}$ |
| **+ ReScore** | $0.20_{\pm0.03}^{+5\%}$ | $\mathbf{0.24}_{\pm0.05}^{+17\%}$ | $79.8_{\pm0.3}^{+1\%}$ | $349.2_{\pm49.6}^{+9\%}$ | $0.11_{\pm0.02}^{+0\%}$ | $0.24_{\pm0.10}^{+5\%}$ | $170.8_{\pm4.0}^{+0\%}$ | $548.0_{\pm91.4}^{+0\%}$ |

