# OpenReview forum: "Boosting Causal Discovery via Adaptive Sample Reweighting"
_ICLR.cc/2023/Conference — ICLR 2023 poster_

### Official Review · Reviewer_m1MC · 2022-10-18

**Confidence:** 4
**Correctness:** 3
**Technical Novelty And Significance:** 4
**Empirical Novelty And Significance:** 3
**Recommendation:** 8

**Clarity, Quality, Novelty And Reproducibility:**

The paper is very clearly written, and easy to follow.  As far as I am aware this approach is novel.  The simulation and real data results suggest that the method works well, and there is associated code that can (presumably) reproduce the authors' results.

**Strength And Weaknesses:**

The method seems useful and clearly improves performance in simulations, as well as on the well-known Sachs data.

The weaknesses are:
1. It is computationally more expensive - this is presumably inevitable given that it involves running the main method several times.
2. No mention is made of the possibility of spurious outliers.  These are generally dealt with in statistics by _down_-weighting, not up-weighting.


**Summary Of The Paper:**

This paper provides a method for improving the robustness of score-based DAG learning methods to heterogeneous noise.  It works by iteratively up-reweighting the poorly fitted observations and then rerunning the base method.


**Summary Of The Review:**

This seems like a nice contribution to the literature, allowing for more accurate learning of DAG structure with score-based algorithms.  Please do ensure you take into account my comments below in your revision.

### Comments

1. What happens if the observations are just statistical outliers, and don't really contain any useful information?  I could easily imagine just a few observations totally disrupting your algorithm in this context, because they would presumably be massively up-weighted and might overwhelm your fitting algorithm.

2. You say that you can fit your method to 'any score-based causal discovery method'.  This is not correct - first, and most obviously, you need the individual data; second the data must be assumed to be independent; and third it is not clear to me whether the method would work for other classes of graphs.

3. "However, it is no doubt that integrating a strong backbone DAG learner with ReScore is the only way to truly advance in this field."  This statement is much too strong; it's not at all clear that another approach might work just as well as yours, or even better!

4. In the proof of Theorem 1, you write down the conditional variance formula at the bottom of page 15 but you implicitly assume that $\mathbb{E} [(X^TWX)^{-1} X^TW N^j \mid X] = 0$.  This is fine (since it's used on the previous line), but you need to state that this is what you're doing, or the proof is hard to follow.  I would suggest giving the conditional variance formula before you start and then noting that this second term is zero.

### Typos/Minor Points

 - page 4: "combinatorial optimizaiton" $\to$ "combinatorial optimization";
 - page 5: "hard less-fitted samples" $\to$ "less well-fitted samples";
 - page 6: "negative llog-likelihood"
 - page 9: "In sheer contrast" $\to$ "In stark contrast"
 - page 15: "one is easy to obtain that" $\to$ "one can easily obtain that"
 - below (9) "of that term" - which term? The one from (8), presumably!
 - page 16: " 'case a.'. " should be just " 'case a.' " and the open quote should be inverted.
 - just below "standard Gaussian distribution": not really, since the variances are not all 1.
 - below (10) "equivalent to minimize the first term": I think you mean the second term.

---

> ### Author Response · Authors · 2022-11-18
> **Response for Reviewer m1MC - 1**
>
> Thanks so much for your time and positive feedback! To address your concerns, we present the point-to-point responses as follows.
> We have carefully revised our paper, taking all your feedback into account.
>
> >**Weakness 2 + Comment 1: Sensitivity to Outliers** - "No mention is made of the possibility of spurious outliers. These are generally dealt with in statistics by down-weighting, not up-weighting." "What happens if the observations are just statistical outliers, and don't really contain any useful information? I could easily imagine just a few observations totally disrupting your algorithm in this context, because they would presumably be massively up-weighted and might overwhelm your fitting algorithm."
>
> We appreciate the valuable comment. This insight sheds light on the need to rethink our method. Hence, in the revision, we **conducted an additional experiment** to empirically evaluate the robustness of ReScore against spurious outliers. Here we introduce this experiment from three aspects: motivations, simulations, and results.
>
> **Motivations.**
> A basic assumption of ReScore is that no pure noise outliers are involved in the training process.
> Otherwise, the DAG learner might get overwhelmed by arbitrarily up-weighting less well-fitted samples, in this case, pure noise data.
> The good news is that the constraint of the cutoff threshold $\tau \in \mathcal{C}(\tau) = \{\mathbf{w}: 0 < \frac{\tau}{n} \leq w_1, \dots, w_n \leq \frac{1}{\tau n}, \sum_{i=1}^{n} w_i = 1\}$ prevents over-exploitation of pure noise samples, which further strengthens ReScore's ability to withstand outliers.
>
> **Simulations.**
> We produce $p_{corrupt}$ percentage pure noise samples in nonlinear settings ($n=2000$, $d = 20$, ER2), where those noise samples are generated from a different structural causal model.
> We select a broad range of $p_{corrupt} = \{0, 0.01, 0.02, 0.05, 0.08, 0.1, 0.2, 0.3, 0.5\}$.
>
>
> **Table 1: SHD for $p_{corrupt}$ percentage noise samples.**
> |                                 | 0    | 0.01 | 0.02 | 0.05 | 0.08 | 0.1  | 0.2  | 0.3  | 0.5  |
> | ------------------------------- | ---- | ---- | ---- | ---- | ---- | ---- | ---- | ---- | ---- |
> | NOTEARS-MLP                     | 14.9 | 15.2 | 15.3 | 18.9 | 19.8 | 21.3 | 23.9 | 23.8 | 28.3 |
> | + ReScore ($\tau \rightarrow 0$ ) | 13.8 | 14.2 | 15.0 | 18.3 | 19.5 | 20.7 | 24.0 | 24.4 | 29.3 |
> | + ReScore (Optimal $\tau$)        | 13.7 | 14.1 | 15.0 | 18.1 | 19.2 | 19.9 | 21.9 | 24.0 | 28.9 |
> | Imp. %                          | +8%  | +7%  | +2%  | +4%  | +3%  | +7%  | +8%  | -1%  | -2%  |
>
> **Results.**
> Table 1 reports the comparison of performance in NOTEARS-MLP and two ReScore methods (no cut-off threshold and optimal $\tau$) when encountering pure noise data.
> Imp.\% measures the relative improvements of ReScore (Optimal $\tau$) over the backbone NOTEARS-MLP.
> We observe that ReScore (Optimal $\tau$) consistently yields remarkable improvements compared with NOTEARS-MLP in the case that less than 20\% of samples are corrupted.
> These results demonstrate the robustness of ReScore when handling data that contains a small proportion of pure noise data.
> Surprisingly, when the cutoff threshold $\tau$ is set to be close to 0, the ReScore can still achieve relative gains over the baseline when less than 10\% of the samples are pure noise.
> Although it is more sensitive to noise samples than the optimum cutoff threshold $\tau$.
> These surprising findings support the effectiveness of adaptive weights and show the potential of ReScore.
>
> >**Comment 2:** - "You say that you can fit your method to 'any score-based causal discovery method'. This is not correct - first, and most obviously, you need the individual data; second the data must be assumed to be independent; and third it is not clear to me whether the method would work for other classes of graphs."
>
> Thanks for your insightful comments. We agree that it is necessary to clarify the scope of our claim, especially for the implicit assumption of ReScore. We carefully **revised our paper** for a more precise claim of applicability. Moreover, in the revision, we **added GES [1]** as our new baseline on the Sachs dataset, which is a discrete score-based causal discovery and scores undirected graphs within an equivalence class. However, sufficiently demonstrating the effectiveness of ReScore in discrete score-based methods requires extending our approach to new baselines and datasets. Due to the limited time of the rebuttal period, we leave the extension of ReScore to other classes of graphs for future work.

---

> > ### Author Response · Authors · 2022-11-18
> > **Response for Reviewer m1MC - 2**
> >
> > GES scores all possible single-edge additions and deletions from all DAGs contained within an equivalence class, where the Bayesian Information Criterion (BIC) is used as the score function.
> > For the bilevel optimization procedure of ReScore, obviously, the main difficulty is in the outer loop, i.e., how to evaluate adding or deleting one edge in a weighted manner.
> > In GES, the specific score is calculated based on the average estimator of Fisher information, which we modify into an unbiased weighted estimator of Fisher information [2,3].
> >
> > To further demonstrate the effectiveness of GES+ReScore, we show the results in Sachs below. Clearly, ReScore strengthens the GES in terms of SHD, TPR, and FDR by a large margin. More detailed comparisons and analyses can be found in our revision paper.
> >
> > **Table 2: GES and GES+ReScore Performance comparison on Sachs**
> > |               | TPR $\uparrow$ | FDR $\downarrow$ | SHD $\downarrow$ | SID $\downarrow$ | #Predicted Edges |
> > | :-----------: | :------------: | :--------------: | :--------------: | :--------------: | :--------------: |
> > |      GES      |     0.294      |      0.853       |        31        |        54        |        34        |
> > | GES + ReScore |     0.588      |      0.722       |      **28**      |        50        |        36        |
> >
> >
> > [1] Optimal structure identification with greedy search.
> >
> > [2] Fisher information in weighted distributions.
> >
> > [3] Extension of covariance selection mathematics.
> >
> > >**Comment 3:** - ""However, it is no doubt that integrating a strong backbone DAG learner with ReScore is the only way to truly advance in this field." This statement is much too strong; it's not at all clear that another approach might work just as well as yours, or even better!"
> >
> > Thanks for pointing out this too-strong statement. We **revised this part** of our paper.
> >
> > >**Comment 4:** - "In the proof of Theorem 1, you write down the conditional variance formula at the bottom of page 15 but you implicitly assume that $E[(X^TWX)^{−1} X^TWN^j∣X] = 0$. This is fine (since it's used on the previous line), but you need to state that this is what you're doing, or the proof is hard to follow. I would suggest giving the conditional variance formula before you start and then noting that this second term is zero."
> >
> > Thanks for your thorough comment. Following your suggestion, we **added conditional variance formula** and **modified our proof** accordingly.
> >
> > >**Comment 5:** - Typos/Minor Points
> >
> > We appreciate the reviewer for pointing out those typos. We thoroughly proofread our paper. Thanks again for your time and in-depth suggestions.

---

### Official Review · Reviewer_x5dL · 2022-10-24

**Confidence:** 3
**Correctness:** 3
**Technical Novelty And Significance:** 3
**Empirical Novelty And Significance:** 2
**Recommendation:** 6

**Clarity, Quality, Novelty And Reproducibility:**

Clarity and Quality:
The paper is well organized and presented. It provides both theoretical guarantee and practical evaluation.

Novelty:
The sample reweighting trick to improve causal discovery performance is novel, as far as I know.



**Strength And Weaknesses:**

Strength:

1. The paper is well presented.
2. The proposed sample reweighting method can help causal discovery in heterogeneous data by auto-learnable adaptive weights.
3. The proposed sample reweighting trick is simple and is flexible to combine with any score-based continuous optimization method in causal discovery.

Weakness:

1. It seems after adding the sample reweighting step, the reported empirical performance, especially FDR, instead gets worse in some cases. Is it because bilevel optimization is harder?

2. With the sample reweighting step, the algorithm is computationally much more expensive.

3. It would be more intuitive to understand the benefit of reweighting if the authors could provide illustrative examples.

**Summary Of The Paper:**

This paper proposes a simple model-agnostic framework to boost causal discovery performance by dynamically learning the adaptive weights for the score function. In particular, the proposed method leverages the bilevel optimization scheme to alternatively train a standard DAG learner first and then reweight the samples that the DAG learner fails to fit well.

**Summary Of The Review:**

The paper is well organized and presented. The sample reweighting trick to improve causal discovery performance is novel, as far as I know. It is also simple and flexible to adapt to other continuous optimization methods.

---

> ### Author Response · Authors · 2022-11-18
> **Response to Reviewer x5dL**
>
> We thank the reviewer for the thorough and valuable feedback. To address your concerns, we present the point-to-point responses as follows.
>
> >**Comment 1:** - "It seems after adding the sample reweighting step, the reported empirical performance, especially FDR, instead gets worse in some cases. Is it because bilevel optimization is harder?"
>
> Thank you very much for your comprehensive analysis of ReScore's results.
> To be honest, we are not entirely certain of the reason for some cases' worse FDR.
> However, we completely agree with you that bilevel optimization is more difficult, and attaining universal improvements in causal discovery with respect to all metrics is challenging.
> ReScore may, in some cases, amplify the detrimental effects of noise data, which, in our opinion, is one of the reasons making FDR worse.
> Thanks again for bringing up this question; we will continue to rethink and explore it.
>
> >**Comment 2:** - "With the sample reweighting step, the algorithm is computationally much more expensive."
>
> Thanks. We agree that ReScore is computationally more expensive. However, as shown in Table 7, we would like to underline that ReScore only adds an acceptable amount of computational complexity. Developing a more effective and efficient reweighting method is left for future research.
>
> >**Comment #:** - "It would be more intuitive to understand the benefit of reweighting if the authors could provide illustrative examples."
>
> We appreciate your constructive comments. In the revision, we **added a new illustrative example** in Appendix. Here we introduce this example from three aspects: motivations, simulations, and results.
>
> **Motivations.**
> To fully comprehend the benefits of reweighting, two research hypotheses need to be verified.
> First, we must determine the validity of the fundamental understanding of ReScore, which states that real-world datasets inevitably include samples of varying importance.
> In other words, there are many informative samples that come from disadvantaged groups in real-world scenarios.
> Additionally, we must confirm that the adaptive weights learned by ReScore faithfully reflect the sample importance, i.e., less well-fitted samples typically come from disadvantaged groups, which are more important than those well-fitted samples.
>
> The real-world Sachs dataset naturally contains nine groups, where each group corresponds with a different experimental condition.
> We first rank the importance of each group in Sachs by using the average weights for each group learned by ReScore as the criterion.
> Then we eliminate 500 randomly selected samples from one specific group, perform NOTEARS-MLP, and show its DAG accuracy inferred from the remaining samples.
> Note that the sample size in each group, which ranges from 700 to 900, is fairly balanced.
>
> **Table 1: Performance comparison for removing samples in different groups**
> | Group Index   | 3     | 5      | 7      | 1      | 2      | 6      | 4      | 8      | 0      |
> | ------------- | ----- | ------ | ------ | ------ | ------ | ------ | ------ | ------ | ------ |
> | Avg. ranking  | 578.4 | 2856.7 | 3368.1 | 3877.0 | 3949.4 | 4549.4 | 4573.2 | 4590.6 | 4910.1 |
> | SHD w/o group | 16    | 16     | 17     | 16     | 16     | 17     | 17     | 19     | 19     |
> | TPR w/o group | 0.529 | 0.412  | 0.412  | 0.412  | 0.412  | 0.412  | 0.412  | 0.353  | 0.294  |
>
> **Results.**
> Table 1 clearly shows a declining trend for SHD and TPR metrics as the significance of deleting groups grows.
> Specifically, removing samples from disadvantaged groups such as Groups 8 and 0, which have the highest average weights, will significantly influence the DAG learning quality.
> In contrast, SHD and TPR of NOTEARS-MLP can even be maintained or slightly decreased by excluding the samples from groups with relatively low average weights.
> This illustrates that samples of different importance are naturally present in real-world datasets, and ReScore is capable of successfully extracting this importance.

---

### Official Review · Reviewer_Q3NX · 2022-10-24

**Confidence:** 4
**Correctness:** 4
**Technical Novelty And Significance:** 3
**Empirical Novelty And Significance:** 4
**Recommendation:** 8

**Clarity, Quality, Novelty And Reproducibility:**

The structure of the paper is very clear, and overall the paper is easy to follow. Exceptions are detailed above.

There are problems with the theoretical results, detailed above.

The proposed method is novel to the best of my knowledge.

I see no problems with reproducibility.




**Strength And Weaknesses:**

Strengths:

* The exposition of the method is quite clear.

* The proposed framework is very flexible, as it can be used on top of many other score-based methods.

Weaknesses / questions about things that are unclear:

* A main concern is that *the paper claims improvements for any score-based method, while this claim is only supported for the case of differentiable score-based methods*. In the experiments, the proposed framework is applied only with differentiable methods. Also, the framework is motivated from observed shortcomings of differentiable methods (bullets page 1&2). It is possible that the proposed framework also gives improvement with discrete score-based methods, but such a claim is not supported by the paper. The scope of the claim needs to be made clear, starting in the abstract.

* Can you explain an example of how single datapoints may provide "crucial causal information", or how without reweighting there is  "overfitting" (both page 2)? I think figure 1 starts to demonstrate such issues, but does not explain these two points.

* In Theorem 1, what does it mean to say "asymptotically", when the vector of weights is held fixed (and so its length can't increase to accommodate more datapoints)?

* The argmax in (6) will often equal the empty set, because the set $\mathbb{C}(\tau)$ is open (so there is a supremum, but no maximum). Replace the inequalities between the $w$'s and the $\tau$-terms by $\leq$ to make the set closed.

* The statement and proof of Theorem 2 are incorrect. The proof can only show that $w_i^* \geq w_j^*$. Because in the proof, when supposing the contradiction, if $w_i^* = w_j^*$, there is no such $\epsilon$. (BTW, the $\mathbb{C}$ should be $\mathbb{C}(\tau)$. Also, you don't need this set to be open.) But this substantially weakens the statement of the theorem: it now leaves open the possibility that all weights are set equal.

* Theorem 2 seems to suggest that the weights are increasing as a function of the losses (over the different datapoints). Can you explain why this is not what we see in Figure 3? (A possible reason might be related to the text "in the optimization phase" in Theorem 2. It is not  clear exactly what point of the bilevel optimization algorithm this refers to.)

* Synthetic data experiment, bottom page 8: what does it mean that the "noise scale is flipped"?

* Sachs data: This dataset contains (I think) 14 different interventional regimes, which are pooled together in this experiment. Pooling is not the most effective way to deal with these data (see eg Mooij, Magliacane and Claassen, 2020). As such, I disagree with the conclusions that this experiment "highlight[s] the ineffectiveness of score-based causal discovery methods when dealing with real-world data" and that "integrating a strong backbone DAG learner with ReScore is the only way to truly advance in this field". Further, due to these regimes, this experiment should be listed under "heterogenous data".

Minor comments (not necessarily part of decision assessment):

* I am not familiar with the terminology "disadvantaged" domain/data point; I think you mean "underrepresented"?

* (page 5) "Initially, ... are assumed": This sentence is unclear. I think you want to say that a *family* of functions and a *family* of noise distributions need to be chosen. The use of the word "specific" confuses me though.

* For the experiments with heterogenous data, you could consider comparing to a method that uses non-adaptive weights, set inversely proportional to the group sizes.

* The manuscript contains many grammatical mistakes, as well as places where the wrong word was chosen (eg "alternative(ly)" when "alternating(ly)" is meant; "phrases" when "phases" is meant; ...). Please thoroughly proofread the manuscript and correct these errors.

**Summary Of The Paper:**

The paper presents a model-agnostic framework that improves pre-existing score-based models for causal discovery. This is done by adapting the score weights for each sample while training the backbone model, thus alternating between training the backbone model and updating the weights.
The weights are updated in such a way as to give a larger impact to the less-fitted samples, which might be more informative to the backbone model. This framework is claimed to have a particularly large impact when dealing with heterogeneous data (even in cases when we do not know what samples come from which cohort).

**Summary Of The Review:**

The proposed method is novel as far as I know, and the experimental results suggest it often yields impressive performance improvements. Because of some issues with the current manuscript, I am currently recommending rejection, but if these issues can be addressed, I expect to recommend acceptance.

---

> ### Author Response · Authors · 2022-11-18
> **Response to Reviewer Q3NX - 1**
>
> We sincerely appreciate your constructive and thorough comments, some of which are fundamental and in-depth suggestions that help us greatly improve our paper. Following your suggestions, we have thoroughly revised our paper, taking all your comments into account.
> If you have additional concerns, we would be pleased to discuss them with you.
>
> > **Comment 1: Scope of Claim** - "A main concern is that the paper claims improvements for any score-based method, while this claim is only supported for the case of differentiable score-based methods. In the experiments, the proposed framework is applied only with differentiable methods. Also, the framework is motivated from observed shortcomings of differentiable methods (bullets page 1&2). It is possible that the proposed framework also gives improvement with discrete score-based methods, but such a claim is not supported by the paper. The scope of the claim needs to be made clear, starting in the abstract."
>
> We appreciate the reviewer bringing this comment up, and we concur that the claim's scope needs to be made clearer. Following your suggestions, we thoroughly **revised our paper**, emphasizing the major focus of our ReScore on differentiable score-based causal discovery right from the title. We **changed our paper** from "Boosting Causal Discovery" to "Boosting Differentiable Causal Discovery".
>
> > **Comment 2: Illustrative Examples** - "Can you explain an example of how single datapoints may provide "crucial causal information", or how without reweighting there is "overfitting" (both page 2)? I think figure 1 starts to demonstrate such issues, but does not explain these two points."
>
> We value your insightful suggestions. To address your concern, we first **revised the introduction** part of our paper, then **conducted an additional illustrative example** to further clarify the motivation behind ReScore.
>
> The illustrative example should highlight two points and better be a real-world dataset. 1. "a small number of informative samples might contain crucial causation information". 2. "ReScore alleviates the overfitting issue of easier-to-fit spurious edges."
> For the first point, the reviewer grabs the fundamental motivation of ReScore, namely the fact that real-world datasets inevitably contain samples of varying importance, with some informative samples that come from disadvantaged groups being more important than others.
> For the second point, the illustrative example needs to clarify how the averaging score function causes overexploitation of easier-to-fit samples, and further learning of spurious edges.
> The second point is supported by ReScore's higher TPR and lower FDR when compared to its corresponding baselines in the Sachs dataset.
>
> The real-world Sachs dataset naturally contains nine groups (thanks for your information! ), where each group corresponds with a different experimental condition.
> We first rank the importance of each group in Sachs by using the average weights for each group learned by ReScore as the criterion.
> Then we eliminate 500 randomly selected samples from one specific group, perform NOTEARS-MLP, and show the DAG accuracy inferred from the remaining samples.
> Note that the sample size in each group, which ranges from 700 to 900, is fairly balanced.
> The results for NOTEARS-MLP in the full Sachs data set are 16 for SHD and 0.412 for TPR.
>
> **Table 1: Performance comparison for removing samples in different groups**
> | Group Index   | 3     | 5      | 7      | 1      | 2      | 6      | 4      | 8      | 0      |
> | ------------- | ----- | ------ | ------ | ------ | ------ | ------ | ------ | ------ | ------ |
> | Avg. ranking  | 578.4 | 2856.7 | 3368.1 | 3877.0 | 3949.4 | 4549.4 | 4573.2 | 4590.6 | 4910.1 |
> | SHD w/o group | 16    | 16     | 17     | 16     | 16     | 17     | 17     | 19     | 19     |
> | TPR w/o group | 0.529 | 0.412  | 0.412  | 0.412  | 0.412  | 0.412  | 0.412  | 0.353  | 0.294  |
>
> Table 1 clearly shows a declining trend in terms of SHD and TPR metrics as the significance of deleting groups grows.
> Specifically, removing samples from disadvantaged groups such as groups 8 and 0, which have the highest average weights, will significantly influence the DAG learning quality.
> In contrast, the SHD of NOTEARS-MLP can even be maintained or slightly decreased by excluding the samples from groups with relatively low average weights.
> This illustrates that samples of different importance are naturally present in real-world datasets, and ReScore is capable of successfully extracting this importance.
> Moreover, the TPR of removing 500 samples from Group 3 is 0.529, which is higher than the 0.412 for using full datasets from Sachs, demonstrating the overfitting of NOTEARS-MLP to easier-to-fit samples, and further learning spurious edges.

---

> > ### Author Response · Authors · 2022-11-18
> > **Response to Reviewer Q3NX - 2**
> >
> > > **Comment 3:** - "In Theorem 1, what does it mean to say "asymptotically", when the vector of weights is held fixed (and so its length can't increase to accommodate more datapoints)?"
> >
> > Thanks. We have **restated the "asymptotically" in Theorem 1** for better comprehension. Please check whether it is satisfactory.
> > Given a data set, the number of observations n is fixed, and so are the weights (since we consider weights at the population level, which can be treated as a parameter). However, the "asymptotically" or the "convergence" in our new statement is considered with the increasing of the data size n. The limit property in Theorem 1 is considered a convergence of random variables in probability theory as n goes to infinity.
> >
> > > **Comment 4:** - "The argmax in (6) will often equal the empty set, because the set $\mathcal{C}(\tau)$ is open (so there is a supremum, but no maximum). Replace the inequalities between the $w$'s and the $\tau$-terms by $≤$ to make the set closed."
> >
> > Thanks for pointing out the issue. We **modified our statement** in our revision.
> >
> > > **Comment 5:** - "The statement and proof of Theorem 2 are incorrect. The proof can only show that $w_i^∗ ≥ w_j^∗$. Because in the proof, when supposing the contradiction, if $w_i^∗ = w_j^∗$ , there is no such $\epsilon$ . (BTW, the $\mathcal{C}$ should be $\mathcal{C}(\tau)$. Also, you don't need this set to be open.) But this substantially weakens the statement of the theorem: it now leaves open the possibility that all weights are set equal."
> >
> > We appreciate your detailed comments. We agree that if we modify the constraint set to be closed, then there exists the issue of contradiction. To address your concern, we **updated our Theorem 2** from strictly “$>$” to “$\geq$” and then **proved the modified theorem 2**. Fortunately, we can show that the equal sign of weights can only hold at the boundary, i.e. $ w^*_i = w^*_j = \frac{\tau}{n} $ or $ w^*_i = w^*_j = \frac{1}{\tau n} $. Moreover, due to the constraint of the sum of weights, i.e., the sum of all weights equals 1, there is no chance for all weights to be equal.
> > More detailed information can be found in Appendix.
> >
> > > **Comment 6:** - "Theorem 2 seems to suggest that the weights are increasing as a function of the losses (over the different datapoints). Can you explain why this is not what we see in Figure 3? (A possible reason might be related to the text "in the optimization phase" in Theorem 2. It is not clear exactly what point of the bilevel optimization algorithm this refers to.)"
> >
> > Thanks for your insightful question. To resolve your concerns, we first **empirically demonstrated** the correctness of Theorem 2. Then we **provided the detailed algorithm** of ReScore in the Appendix. Lastly, we **updated section 4.2** to clearly describe the findings in Figure 3.
> >
> > We calculate both Pearson and Spearman correlations between the weights and losses in the optimization phase. The results are displayed in Table 2. In accordance with the notations in the Algorithm 1, $K_{inner}$ refers to the maximum iterations in the inner loop where the adaptive weights are learned, and $k_1$ denotes the number of iterations in the outer loop.
> > We observe that as the iteration for outer loop increases, the correlation grows as we expect.
> > Moreover, if we set a larger $K_{inner}$, the correlation raises.
> > This illustrates the learning process of adaptive weights and empirically demonstrates the oracle property in Theorem 2, which states that the weights increase as a function of the losses as $k_1 \rightarrow \infty$ and $K_{inner}$ becomes sufficiently large.
> >
> > **Table 2: The correlation between weights and losses in the optimization phase**
> > |                 |          | $k_1=0$ | $k_1=1$ | $k_1=2$ | $k_1=3$ | $k_1=4$ |
> > | :-------------: | :------: | :-----: | :-----: | :-----: | :-----: | :-----: |
> > | $K_{inner} = 5$ |  Pearson  |  0.366  |  0.485  |  0.571  |  0.647  |  0.692  |
> > | $K_{inner} = 5$ | Spearman |  0.326  |  0.459  |  0.563  |  0.648  |  0.705  |
> > | $K_{inner} = 8$ |  Pearson  |  0.606  |  0.717  |  0.828  |  0.859  |  0.865  |
> > | $K_{inner} = 8$ | Spearman |  0.524  |  0.654  |  0.806  |  0.841  |  0.867  |
> >
> > This learning process of adaptive weights also explains why the findings in Figure 3 appear to violate Theorem 2.
> > The limited number of inner loops prevents the adaptive weights from optimizing to the best value in each outer loop.
> > As a result, samples from the disadvantaged group that had relatively high losses in the initial iteration are selected and cumulatively assigned bigger weights.
> >
> > > **Comment 7:** - "Synthetic data experiment, bottom page 8: what does it mean that the "noise scale is flipped"?"
> >
> > The scales of noise variables are flipped means that for the identical noise variable $N_i$, if $N_i \sim \mathcal{N}(0,1)$ in the disadvantaged group, then $N_i \sim \mathcal{N}(0,0.1)$ in the dominant group.
> > We also **updated Section 4.2.1** for clarity and readability.

---

> > > ### Author Response · Authors · 2022-11-18
> > > **Response to Reviewer Q3NX - 3**
> > >
> > > > **Comment 8:** - "Sachs data: This dataset contains (I think) 14 different interventional regimes, which are pooled together in this experiment. Pooling is not the most effective way to deal with these data (see eg Mooij, Magliacane and Claassen, 2020). As such, I disagree with the conclusions that this experiment "highlight[s] the ineffectiveness of score-based causal discovery methods when dealing with real-world data" and that "integrating a strong backbone DAG learner with ReScore is the only way to truly advance in this field". Further, due to these regimes, this experiment should be listed under "heterogenous data"."
> > >
> > > Thanks so much for bringing this recent work to us. Your important information and in-depth knowledge of the Sachs dataset also motivate us to develop a new illustrative example (depicted in Comment 2). Following your suggestions, we have **re-organized the sections** and shifted the Sachs results into the heterogeneous part. We also **updated the analysis** of Sachs results based on your comments.
> > >
> > > > **Minor Comment 9:** - "I am not familiar with the terminology "disadvantaged" domain/data point; I think you mean "underrepresented"?"
> > >
> > > Yes. the terminology "disadvantaged" is used in the tasks of long-tail distribution and domain adaptation.
> > >
> > > > **Minor Comment 10:** - "(page 5) "Initially, ... are assumed": This sentence is unclear. I think you want to say that a family of functions and a family of noise distributions need to be chosen. The use of the word "specific" confuses me though."
> > >
> > > We appreciate that the reviewer brought up this confusing point. The sentence has been modified.
> > >
> > > > **Minor Comment 11:** - "For the experiments with heterogenous data, you could consider comparing to a method that uses nonadaptive weights, set inversely proportional to the group sizes."
> > >
> > > Thanks for your valuable comments. We have **compared an additional baseline** (a non-adaptive reweighting method) on heterogeneous data, in which sample weights are inversely proportional to group sizes. The detailed results can be found in our revision. Briefly speaking, the poor performance of the non-adaptive baseline helps us clearly highlight the significance of the adaptive property and reveal the advantages of ReScore. We appreciate your ideas.
> > >
> > > > **Minor Comment 12:** - "The manuscript contains many grammatical mistakes, as well as places where the wrong word was chosen (eg "alternative(ly)" when "alternating(ly)" is meant; "phrases" when "phases" is meant; ...). Please thoroughly proofread the manuscript and correct these errors."
> > >
> > > We thank the reviewer for pointing out these typos. We carefully proofread the manuscript and sincerely hope that you will find the revision satisfactory. We appreciate your time and insightful comments.

---

> > > > ### Comment · Reviewer_Q3NX · 2022-11-30
> > > > **Thanks**
> > > >
> > > > Dear authors, thank you for answering my comments so thoroughly. I find that the revised paper addresses the concerns I had, and I have raised my score accordingly.
> > > >
> > > > One final comment: there was a cut-off sentence in the "Simulations" paragraph of the revised manuscript.

---

### Official Review · Reviewer_ueyq · 2022-10-28

**Confidence:** 4
**Correctness:** 4
**Technical Novelty And Significance:** 3
**Empirical Novelty And Significance:** 2
**Recommendation:** 6

**Clarity, Quality, Novelty And Reproducibility:**

Overall, I think the authors have written a clear and well organized paper. Outside of the issues that I raised I think the quality is quite good and the novelty is quite high, in my view.

**Strength And Weaknesses:**

Strengths:
* The relative frailty of structure learning algorithms is a major roadblock to practical usage, so this task is very well motivated.
* The authors present an algorithm which is simple and intuitive
* Empirical results are very compelling

Weaknesses:
* The authors reweight samples with the intuition that these correspond to spurious edges, however it is not entirely clear to me why this should be limited to one reweighting step. Can the authors give some sort of intuition on why only one boosting step is used? What would happen if multiple reweighting steps were employed?
* Theorem one is incredibly limited in scope. I don't think this necessarily a problem but the authors should reconcile their claim that the approach is applicable to any score base learner with the assumptions that restrict the space of models and algorithms substantially.
* The approach deals with recovering directed graphs, but the authors frame in terms of causal discovery. In the case of causal discovery, only an equivalence class is recovered, not a fully directed graph. Do the authors have suggestions or intuition regarding the modifications necessary to have the algorithm work when an algorithm (e.g., GES) returns an equivalence class after every step?

**Summary Of The Paper:**

This work addresses the problem of errors in structure learning algorithms.  The authors propose to reweight poorly fit samples in order to improve the efficacy of the underlying algorithm. A proof is shown under linear models for specific scoring rules in the asymptotic regime. Empirical results show the proposed method improving state of the art structure learning algorithms.

**Summary Of The Review:**

As I mentioned above, I think this is a well motivated and simple solution to a very compelling problem. I think the idea that the procedure can be applied a wide variety of algorithms is quite compelling. My reservations are listed above in the weaknesses, and is largely in the theoretical underpinnings and some of the details of the proposed approach.

---

> ### Author Response · Authors · 2022-11-18
> **Response to Reviewer ueyq**
>
> We sincerely thank you for your time and valuable comments. Your main suggestions on the scope of our claim and the extension of our algorithm help us refine our paper. To address your concerns, we present the point-to-point response as follows.
>
> > **Comment 1: Multiple Reweighting Steps** — “The authors reweight samples with the intuition that these correspond to spurious edges, however it is not entirely clear to me why this should be limited to one reweighting step. Can the authors give some sort of intuition on why only one boosting step is used? What would happen if multiple reweighting steps were employed?”
>
> We are not entirely sure if we misunderstood your comment regarding multiple reweighting steps. If we have misunderstood, please feel free to point it out so we can further clarify.
>
> We formulate ReScore as a bilevel optimization problem where the adaptive weights are learned and the DAG learner is trained alternately. In other words, ReScore employs multiple reweighting procedures. We wholeheartedly concur with you that multiple reweighting steps could be superior to fixed weights or a one-step reweighting step since they can gradually help the DAG learner evolve.
>
> Thanks for your question. We realize that a detailed description of the ReScore algorithm can be helpful. In light of this, we **added the ReScore algorithm** to Appendix B and **polished Section 3.1** in our revision.
>
> > **Comment 2: Scope of Claim** - "Theorem one is incredibly limited in scope. I don't think this necessarily a problem but the authors should reconcile their claim that the approach is applicable to any score base learner with the assumptions that restrict the space of models and algorithms substantially."
>
> Thanks for pointing out this comment. We agree that the scope of our claim could be better clarified. Following your suggestions, we **revised our paper** and emphasized the restriction of our theorem 1 in Section 3.
>
> > **Comment 3: Applicability to Discrete Score-based Methods** - "The approach deals with recovering directed graphs, but the authors frame in terms of causal discovery. In the case of causal discovery, only an equivalence class is recovered, not a fully directed graph. Do the authors have suggestions or intuition regarding the modifications necessary to have the algorithm work when an algorithm (e.g., GES) returns an equivalence class after every step?"
>
> Thanks so much for your great suggestions! It is indeed a promising direction to extend our ReScore to discrete score-based causal discovery methods. However, sufficiently demonstrating the effectiveness of ReScore in discrete methods (i.e., equivalence class of graph, undirected graph type) requires extending our approach to new baselines and datasets, which we leave for future work. What we did during the rebuttal period is showcasing the applicability of ReScore to GES.
>
> To answer your concern, first, we **revised our paper** to emphasize that our ReScore mainly focuses on differentiable score-based causal discovery methods, where ReScore achieves significant improvements supported by both empirical experiments and theoretical proof.
> Second, due to the limited time of the rebuttal period, we only **added GES** as you suggest, as our new baseline in the Sachs dataset. We briefly introduce the intuition and real data results of GES+ReScore below.
>
> For Greedy Equivalence Search (GES [1]), it scores all possible single-edge additions and deletions from all DAGs contained within an equivalence class, where the Bayesian Information Criterion (BIC) is used as the score function.
> For the bilevel optimization procedure of ReScore, obviously, the main difficulty is in the outer loop, i.e., how to evaluate adding or deleting one edge in a weighted manner.
> In GES, the specific score is calculated based on the average estimator of Fisher information, which we modify into an unbiased weighted estimator of Fisher information [2,3].
>
> To further demonstrate the effectiveness of GES+ReScore, we show the results in Sachs here. Clearly, ReScore strengthens the GES in terms of SHD, TPR, and FDR by a large margin. More detailed comparisons and analyses can be found in our revision paper.
>
> **Table 1: GES and GES+ReScore Performance comparison on Sachs**
> |               | TPR $\uparrow$ | FDR $\downarrow$ | SHD $\downarrow$ | SID $\downarrow$ | #Predicted Edges |
> | :-----------: | :------------: | :--------------: | :--------------: | :--------------: | :--------------: |
> |      GES      |     0.294      |      0.853       |        31        |        54        |        34        |
> | GES + ReScore |     0.588      |      0.722       |      **28**      |        50        |        36        |
>
>
>
> [1] Optimal structure identification with greedy search.
>
> [2] Fisher information in weighted distributions.
>
> [3] Extension of covariance selection mathematics.

---

### Author Response · Authors · 2022-11-30
**General response to reviewers**

We appreciate all the reviewers for their valuable comments and suggestions. They helped us improve our submission and better strengthen our claims. Here we summarize the major updates brought to the revised manuscript:


- **Revised paper & More clarifications on the scope.** To address a shared concern of Reviewers ueyq, Q3NX, and m1MC, we have thoroughly revised our paper, emphasizing the major focus of our ReScore on **differentiable score-based causal discovery** right from the title.

- **Real-world illustrative example.** To better support the motivation of reweighting and address the concern of Reviewers Q3NX and x5dL, we added a new illustrative example in Appendix.

- Following the suggestions of Reviewer Q3NX, we **restated Theorem 1**, **updated the proof of Theorem 2**, **reorganized the experiments section**, **compared additional baselines** on heterogeneous data, **provided the detailed algorithm** in Appendix, and **added an additional experiment** to empirically illustrate Theorem 2.

- **More experiments.** We added an additional experiment that evaluates the robustness of ReScore against spurious outliers to address the concern of Reviewer m1MC.


We have tried our best to address the main concerns raised by reviewers and we hope that these improvements will be taken into consideration. Updates in the revision are highlighted in blue. We also present the point-to-point responses for each reviewer below.

---

### Decision · Program_Chairs · 2023-01-20

**Decision:**

Accept: poster

**Justification For Why Not Higher Score:**

Although the method was demonstrated to be useful, its properties are not totally clear. For instance, after adding the sample reweighting step, the reported empirical performance, especially FDR, actually gets worse in some cases, and the reason is unclear.

**Justification For Why Not Lower Score:**

The proposed sample reweighting method can help causal discovery in heterogeneous data by auto-learnable adaptive weights.

**Metareview: Summary, Strengths And Weaknesses:**

This paper proposes a simple but effective method to boost causal discovery performance by dynamically learning the adaptive weights for the reweighted score function. It works by iteratively up-reweighting the poorly fitted observations and then rerunning the base method. The method seems useful and clearly improves performance on simulated data and the Sachs data. As the price to pay, it is computationally more expensive. Overall, the paper is nicely written, and the proposed method is novel and has been demonstrated to work well across multiple scenarios.

**Note From Pc:**

if the above contains the word "oral" or "spotlight" please see: "oral" presentation means -> notable-top-5% and "spotlight" means -> notable-top-25%. As stated in our emails, we are disassociating presentation type from AC recommendations